# The metal cofactor zinc and interacting membranes modulate SOD1 conformation-aggregation landscape in an in vitro ALS model

Achinta Sannigrahi[1†], Sourav Chowdhury[2†], Bidisha Das[1,3†], Amrita Banerjee[4], Animesh Halder[5], Amaresh Kumar[6], Mohammed Saleem[6], Athi N Naganathan[7], Sanat Karmakar[5], Krishnananda Chattopadhyay[1,3]*

[1]Structural Biology & Bio-Informatics Division, CSIR-Indian Institute of Chemical Biology, Kolkata, India; [2]Chemistry and Chemical Biology, Harvard University, Cambridge, United States; [3]Academy of Scientific and Innovative Research (AcSIR), CSIR- Human Resource development Centre Campus, Ghaziabad, India; [4]Hiralal Mazumdar Memorial College for Women, Kolkata, India; [5]Department of Physics, Jadavpur University, Kolkata, India; [6]School of Biological Sciences, National Institute of Science Education and Research (NISER), Bhubaneswar, India; [7]Department of Biotechnology, Bhupat & Jyoti Mehta School of Biosciences, Indian Institute of Technology Madras, Chennai, India

**Abstract** Aggregation of Cu–Zn superoxide dismutase (SOD1) is implicated in the motor neuron disease, amyotrophic lateral sclerosis (ALS). Although more than 140 disease mutations of SOD1 are available, their stability or aggregation behaviors in membrane environment are not correlated with disease pathophysiology. Here, we use multiple mutational variants of SOD1 to show that the absence of Zn, and not Cu, significantly impacts membrane attachment of SOD1 through two loop regions facilitating aggregation driven by lipid-induced conformational changes. These loop regions influence both the primary (through Cu intake) and the gain of function (through aggregation) of SOD1 presumably through a shared conformational landscape. Combining experimental and theoretical frameworks using representative ALS disease mutants, we develop a 'co-factor derived membrane association model' wherein mutational stress closer to the Zn (but not to the Cu) pocket is responsible for membrane association-mediated toxic aggregation and survival time scale after ALS diagnosis.

*For correspondence:
krish@iicb.res.in

†These authors contributed equally to this work

Competing interests: The authors declare that no competing interests exist.

## Introduction

The aggregation of SOD1 is believed to be one of the chief causative factors behind the lethal motor neuron disease, amyotrophic lateral sclerosis (ALS) (*Shaw and Valentine, 2007*). Although more than 140 SOD1 mutations have been reportedly associated with ALS, there is no correlation between the stability (and aggregation) of these mutations and their disease manifestations. SOD1 aggregation has been investigated extensively in vitro by altering the solution conditions, such as temperature, pH, the presence of metal chelators, and the reduction of disulfide linkages (*Bush, 2002*; *Niwa, 2007*; *Rodriguez et al., 2005*). The results of these studies clearly suggest that the aggregation of SOD1 is heterogeneous containing multiple steps, which is presumably the reason behind the lack of a structural understanding of aggregation processes (*Pasinelli et al., 2004*; *Tomik et al., 2005*).

**eLife digest** Amyotrophic lateral sclerosis, or ALS, is an incurable neurodegenerative disease in which a person slowly loses specialized nerve cells that control voluntary movement. It is not fully understood what causes this fatal disease. However, it is suspected that clumps, or aggregates, of a protein called SOD1 in nerve cells may play a crucial role.

More than 140 mutations in the gene for SOD1 have been linked to ALS, with varying degrees of severity. But it is still unclear how these mutations cause SOD1 aggregation or how different mutations influence the survival rate of the disease. The protein SOD1 contains a copper ion and a zinc ion, and it is possible that mutations that affect how these two ions bind to SOD1 influences the severity of the disease.

To investigate this, Sannigrahi, Chowdhury, Das et al. genetically engineered mutants of the SOD1 protein which each contain only one metal ion. Experiments on these mutated proteins showed that the copper ion is responsible for the protein's role in neutralizing harmful reactive molecules, while the zinc ion stabilizes the protein against aggregation. Sannigrahi et al. found that when the zinc ion was removed, the SOD1 protein attached to a structure inside the cell called the mitochondria and formed toxic aggregates.

Sannigrahi et al. then used these observations to build a computational model that incorporated different mutations that have been previously associated with ALS. The model suggests that mutations close to the site where zinc binds to the SOD1 protein increase disease severity and shorten survival time after diagnosis. This model was then experimentally validated using two disease variants of ALS that have mutations close to the sites where zinc or copper binds.

These findings still need to be tested in animals and humans to see if these mechanisms hold true in a multicellular organism. This discovery could help design new ALS treatments that target the zinc binding site on SOD1 or disrupt the protein's interactions with the mitochondria.

WT SOD1 contains $Cu^{2+}$ (Cu) and $Zn^{2+}$ (Zn) as cofactors. It has been established that Cu is responsible for the primary function of SOD1 (the dismutase activity), and cell membrane acts as a scaffold in the process of Cu transfer to apo-SOD1 (metal free non-functional protein) through a Cu delivery chaperone (CCS) (*Culotta et al., 1997*). Previous studies have found noticeable presence of SOD1 in human serum lipoproteins, mainly in LDL and HDL, hinting at a possible protective role of SOD1 against the lipid peroxidation (*Mondola et al., 2016*). It has also been noted that SOD1 has a physiological propensity to accumulate near the membranes (*Ilieva et al., 2009*) of different cellular compartments, including mitochondria, endoplasmic reticulum (ER), and Golgi apparatus (*Manfredi and Kawamata, 2016*). In addition, computational studies have shown that the electrostatic loop (loop VII, residues 121–142) and Zn-binding loop (loop IV, residues 58–83) promote membrane interaction of apo-SOD1 initiating the aggregation process (*Chng and Strange, 2014*). Membrane binding induced aggregation of SOD1 has also been shown experimentally both in vitro and inside cells (*Hervias et al., 2006*; *Yamanaka et al., 2008*; *Choi et al., 2011*). Inclusions of SOD1 have been detected in the inter-membrane space of mitochondria originating from the spinal cord (*Mondola et al., 2016*).

The above results can be reconciled by suggesting that cell membrane can play crucial roles not only in shaping up the primary function of the protein, but also in defining its aggregation process of generating fibrillar and non-fibrillar aggregates (the gain of function), with the loops IV and VII contributing critically to both processes. We hypothesize that (1) the induction of metal cofactors for the stabilization of loop IV and VII, membrane interaction, and SOD1 aggregation would be some of the crucial elements in defining the overlapping folding-aggregation landscape of SOD1; (2) metal pocket perturbation by mutational stresses (as in disease variants) would modulate membrane association and facilitate aggregation, and (3) the difference in aggregate morphology as a result of differential membrane interaction may contribute to the variation in cellular toxicity observed in ALS.

In this paper, we investigated the above hypothesis by studying how different structural elements (i.e. co-ordination of individual metals, membrane association, and the location of mutations) attenuate the toxic gain of function of SOD1. An effective understanding of the role of individual metals (Cu and Zn) would require studying SOD1 variants containing only one metal (Cu or Zn) in addition

to a variant that contains none. We have therefore prepared an apo (metal free) protein, which serves the latter purpose. For the former, we have generated two single metal containing mutants of SOD1, viz. H121F (only Zn, no Cu) and H72F (only Cu, no Zn), which are situated near the key loop VII (H121F) and loop IV (H72F) at the protein structure (*Figure 1a*).

Using computational analysis based on a statistical mechanical model and detailed in vitro experiments, we propose here a '**Co-factor derived membrane association model'** of SOD1 aggregation and its possible implication in ALS. We demonstrate that differential metal binding and membrane assisted conformational changes can work in concert to attenuate the rate and propensity of aggregation. While apo (no metal) protein and H72F mutant (no Zn) experience strong membrane interaction, the WT (both metals) protein and H121F (no Cu) mutant do not show significant binding. We further find that membrane-induced aggregates of H72F and apo protein showed significantly higher toxicity in terms of cell death and model membrane deformation when compared to WT and H121F mutant. We finally check the validity of this model to ALS using computational and experimental studies. For the computational validation, we show, using 15 ALS disease mutants, that the distance between the mutation site and Zn correlates well with the membrane binding energy and patient survival time after disease diagnosis, while Cu site does not seem to have any prominent role. For the experimental study, we use two well-studied disease mutants (G37R where mutational site is close to the Cu pocket and I113T where mutational site is near Zn pocket) to show that the model accounts well for their membrane binding/aggregation, correlating well with their disease onset phenotypes. This model puts forward a mechanism that Zn pocket destabilization (either by metal content variation or by mutational stress near Zn center) is the driving force behind the toxic gain of function of SOD1 mediated by the process of membrane association.

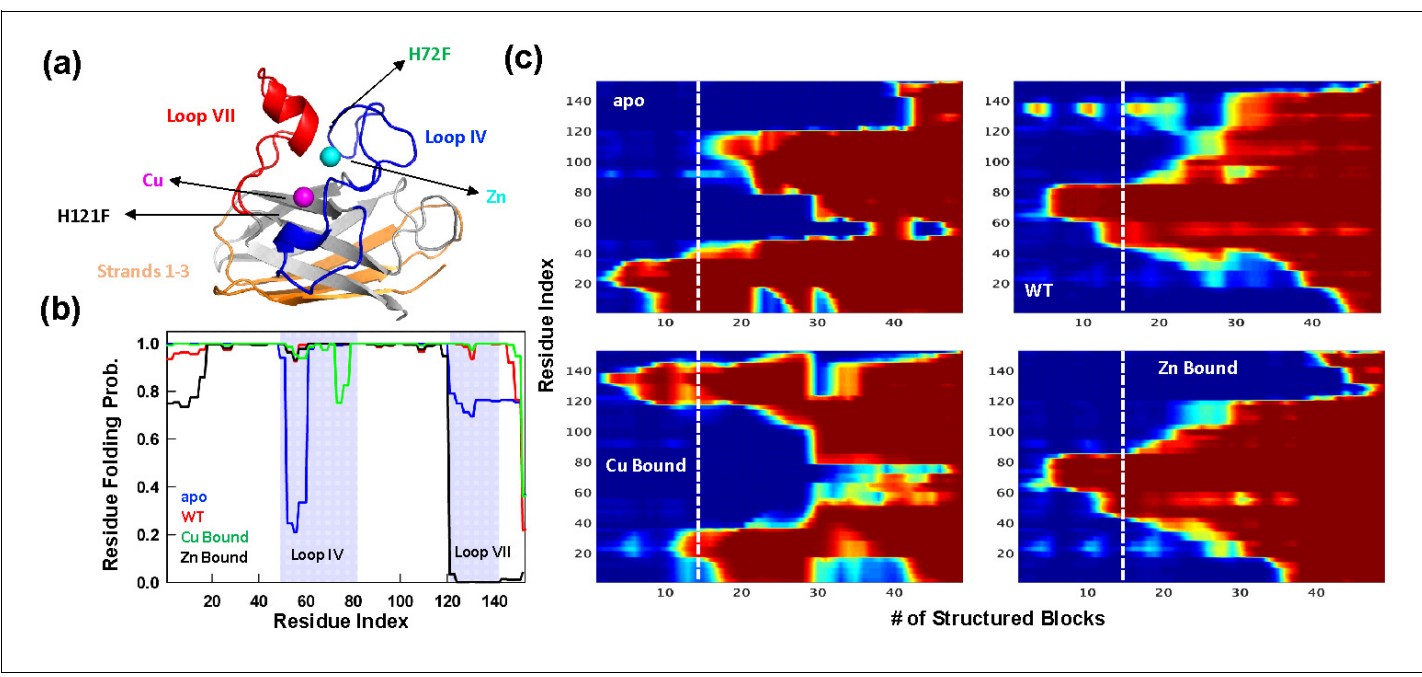

**Figure 1.** Statistical mechanical modeling of SOD1 folding mechanism. (**a**) Cartoon representation of SOD1 monomer highlighting the various structural elements. The positions of the mutation for H121F and H72F have been arrow marked. (**b**) Residue folding probability as a function of residue index for the different variants of SOD1 as predicted by the bWSME model. Note that WT represents the variant in which both the metallic cofactors are bound. (**c**) Average folding probabilities colored in the spectral scale going from 0 (dark blue) to 1 (dark red) as a function of the reaction coordinate, number of structured blocks. The vertical white dashed line signals the parts of the protein that fold first. For example, it can be seen that residues 1–40 fol d early in the apo SOD1 (dark red) when compared to WT where residues 40–80 fold first.

The online version of this article includes the following figure supplement(s) for figure 1:

**Figure supplement 1.** Aggregation prone regions as a function of sequence from AGGRESCAN software (http://bioinf.uab.es/aggrescan/ ).

## Results

### Statistical mechanical modeling of SOD1 folding mechanism hints at aggregation origins

The large size of SOD1 (151 residues) precludes a detailed characterization of the conformational landscape, the role of ions in determining the stability-folding mechanism, and the effect of numerous mutations via all-atom simulation methods. To overcome this challenge and to obtain a simple physical picture of how the energetics of folding is governed by metal ions, we resort to constructing the folding landscape of reduced SOD1 variants through the statistical mechanical Wako–Saitô–Muñoz–Eaton (WSME) model (see Materials and methods for model description and parametrization) (*Wako and Saitô, 1978*; *Muñoz and Eaton, 1999*). Here, we employ the bWSME model where stretches of three consecutive residues are considered as a block (b) that reduces the total number of microstates from 42.7 million to just ~450,000 (*Gopi et al., 2019*).

The model, however, incorporates contributions from van der Waals interactions, simplified solvation, Debye–Hückel electrostatics, excess conformational entropy for disordered residues, and restricted conformational freedom for proline residues (*Naganathan, 2012*; *Rajasekaran et al., 2016*). The predicted average folding path of SOD1 WT (with both Cu and Zn bound) highlights that the folding is initiated around the metal binding regions with early folding of the loop IV (nucleated by Zn) in the unfolded well and aided by flickering structure in the electrostatic loop (loop VII, nucleated by Cu). The rest of the structure coalesces around this initial folding site leading to the native state. This folding mechanism is very similar to that proposed earlier *via* detailed kinetic studies (*Leinartaite et al., 2010*). In the absence of metal ions, the apo variant folds through an alternate pathway wherein the folding is nucleated through the first three strands (residues 1–40) following which the rest of the structure folds, recapitulating the results of single-molecule experiments (*Figure 1*; *Sen Mojumdar et al., 2017*). It is important to note that the first three strands exhibit higher aggregation propensity as predicted from different computational servers (*Figure 1—figure supplement 1–*). Interestingly, the folding mechanism of the Zn-bound SOD1 (with no Cu bound) is similar to that of the WT hinting that Zn coordination promotes proper folding. On the other hand, the folding mechanism of the Cu-bound SOD1 (in the absence of Zn) is similar to that of apo-SOD1 with additional folding probability in the region around the electrostatic loop. Taken together, the statistical modeling highlights how the absence of metals and particularly the absence of Zn (or mutations that affect Zn binding and not Cu binding) alters the folding mechanism by populating partially structured states involving beta strands in the unfolded well thus possibly increasing the chances of aggregation. Importantly, the model provides multiple testable predictions on the differential roles of Zn and Cu, which we address below *via* experiments.

### Cu-deficient H121F behaves like WT SOD1, whereas zn-deficient H72F behaves like apo

We have recently shown that the mutants H121F and H72F contain negligible Cu and Zn, respectively, while the apo protein is completely devoid of metal (*Chowdhury et al., 2019*). We validated this further using atomic absorption spectroscopy (*Table 1*) and activity measurements (*Figure 2—figure supplement 1*). Guanidinium-induced equilibrium unfolding transitions of H121F and H72F were found to be similar (*Figure 2—figure supplement 2*). We used steady-state tryptophan fluorescence, far UV CD, and FTIR spectroscopy to characterize these different proteins. SOD1 is a single tryptophan protein (Trp32), in which the tryptophan residue has been shown to be partially

**Table 1.** Metal contents (Cu and Zn) in WT and other mutants (H121F, H72F, and apo) as obtained from atomic absorption spectroscopy.

| Protein forms | Cu content | Zn content |
|---|---|---|
| WT | 4.5 μM | 3.9 μM |
| H121F | <1 μM | 3.8 μM |
| H72F | 4.1 μM | <1.2 μM |
| apo | <0.5 μM | <0.25 μM |

buried (*Muneeswaran et al., 2014*). The role of Trp32 within the sequence segment (*Lomize et al., 2012*) KVWGSIKGL (*Gohil and Greenberg, 2009*) of high aggregation propensity has been investigated before (*Taylor et al., 2007*). We found that the formation of apo form resulted in a large shift in Trp32 emission maximum (332 nm for WT protein and 350 nm for apo protein) (*Figure 2a*). In contrast, other two mono-metallated variants (H121F and H72F) exhibited fluorescence emission maxima at wavelengths, which were intermediate between the WT and apo proteins (342 nm for H121F and 345 nm for H72F) (*Figure 2a*). Next, we performed acrylamide-quenching experiments to measure the solvent surface exposure of Trp32 for all variants. The values (*Table 2*) of the Stern−Volmer constant ($K_{sv}$) were determined using a straight line fit, as shown in *Figure 2—figure supplement 3*. $K_{sv}$ for WT ($6.8 \pm 0.1$ M$^{-1}$) was significantly lower than that of apo SOD1 ($14.3 \pm 0.1$ M$^{-1}$). Steady-state fluorescence maxima in combination with acrylamide quenching data suggested an appreciable conformational alteration in going from the WT to the apo form. Interestingly, Zn-starved H72F mutant showed higher $K_{sv}$ compared to Cu-starved H121F mutant (*Figure 2—figure supplement 3*, *Table 2*).

Far-UV circular dichroism (CD) spectra for WT and apo protein were in line with earlier observations (*Figure 2—figure supplement 4*; *Banci et al., 2007*). Specifically, we found a slight broadening in the far UV CD spectrum of the apo protein when compared to the WT. In agreement with steady-state fluorescence data, the far UV-CD spectrum of the Zn-deficient H72F protein was found to be similar to the apo variant, while the WT- and Cu-deficient H121F variant displayed similar spectra. We then used FT-IR spectroscopy to complement far-UV CD results and to obtain a preliminary estimate of the secondary structure contents of the protein variants, using amide-I FTIR spectral region. The carbonyl (C = 0) stretching vibrations at amide-I region provides information related to the secondary structure (beta sheet 1633–1638, alpha helix 1649–1656, disorder and turns and loops 1644 and 1665–1672 cm$^{-1}$). The analyses of the FT-IR data were carried out using published method using two steps (*Yang et al., 2015*; *Kong and Yu, 2007*; *Bandyopadhyay et al., 2021*). First, the peak positions were assigned using the double derivatives of the FT-IR data for all protein variants (*Figure 2—figure supplement 5* shows the representative double derivative plot of WT SOD1 in the absence of lipid). The peak positions were selected from the minima of the secondary derivatives of the FT-IR absorbance data. In the second step, selected peak positions thus determined were used for the fitting of the FT-IR raw data using Gaussian distributions analyses. Analysis of the secondary structure of WT protein (*Figure 2b*) showed the presence of 10% alpha helix, 38% beta sheet, and 52% turns and loops including disordered stretches. The percentage of the secondary structure determined from the FT-IR analysis was found to be consistent with the data obtained from the crystal structure (PDB 4BCY with 11% alpha helix, 40% beta sheet, and 49% turns and loops), thus validating our method (*Danielsson et al., 2013*). FT-IR data showed a decrease in beta sheet content (from 38% to 31%) as apo protein (*Figure 2c*) formed. In contrast, the behavior of H72F mutant (*Figure 2—figure supplement 6a*, beta sheet content of 32%) was found to be similar to the apo protein, while H121F mutant (*Figure 2—figure supplement 6b*, beta sheet content of 37%) remained similar to WT protein. The percentage of secondary structure elements of all protein variants are shown in *Figure 2d*.

## Zn-deficient SOD1 shows higher membrane association compared to the cu-deficient and WT proteins

To obtain a preliminary understanding of the possible membrane binding sites of SOD1, we resorted to computational techniques using 'Orientation of protein in membrane' tool (*Lomize et al., 2012*), which predicted weak interaction of WT on membrane surface (*Figure 2—figure supplement 7a*). In contrast, the same calculation predicted higher binding affinity of apo protein with the membrane (*Figure 2e*). When we used ITASSER-modeled structures, the computed values of $\Delta G_{transfer}$ (free energy change of protein transfer from bulk to the membrane) was found to be substantially higher for the apo ($-2.6$ kcal mol$^{-1}$) when compared to the WT protein ($-1.2$ kcal mol$^{-1}$). When we used the crystal structure of the WT protein, $\Delta G_{transfer}$ calculation modeling yielded similar results for the WT protein ($-0.9$ kcal mol$^{-1}$). To probe protein-lipid binding constants experimentally ($K_a$, M$^{-1}$), we used fluorescence correlation spectroscopy (FCS). FCS monitors diffusional and conformational dynamics of fluorescently labeled biomolecules at single-molecule resolution (*Chattopadhyay et al., 2002*). For FCS experiments, we labeled the cysteine residues of all the SOD1 variants using Alexa-488-maleimide. *Figure 2f* shows a schematic diagram of how the labeled proteins and protein-lipids

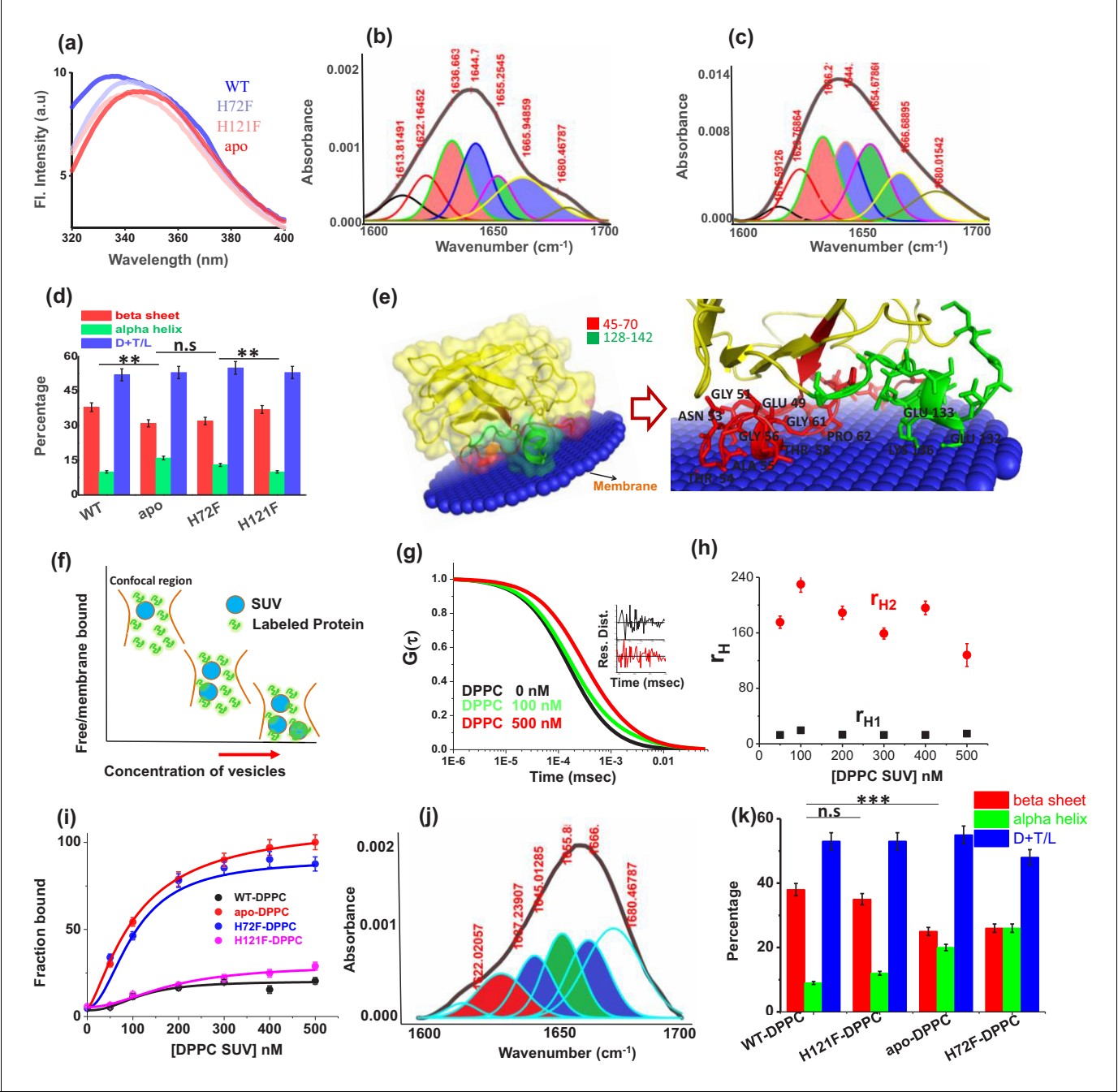

**Figure 2.** Structural characterization of SOD1 mutants and membrane association. (a) Steady-state tryptophan fluorescence spectra of WT, apo, and other two metal mutants (H121F and H72F). The WT displays an emission maximum at 332 nm, whereas the apo variant shows a red-shifted spectrum with the emission maximum at 350 nm. On the other hand, H121F and H72F show emission maxima at intermediate wavelengths. Deconvoluted FTIR spectral signatures of (b) WT and (c) apo. Red contour (~1637 $cm^{-1}$) indicates beta sheet; blue color contour stands for disorder (1644 $cm^{-1}$) and loops and turns (~1667 $cm^{-1}$); green contour represents alpha helical character. All these secondary signatures were obtained by considering the amide-I spectra, which arises due to carbonyl frequency (C = O). (d) Percentage of different secondary structural components in WT, apo, H121F, and H72F are shown in this figure. n.s denotes nonsignificant change, while ** stands for significant change with p-value<0.01. Error bars indicate the standard deviation of the data, which were obtained from triplicate experiments. Here, D + T/L stands for Disorder +Turns/Loops. (e) The membrane association of the apo protein as suggested by the OPM calculations. The membrane association of apo protein through the stretches 45–70 and 128–142 has been evaluated from the calculations. The residues which are involved in binding with membrane (Thr54,58, Ala54, Gly56,61, Pro62, Asn53, Glu49, Lys136, Glu132,133) are mentioned. (f) A schematic representation regarding the membrane binding experiments through FCS which suggests that with increasing concentration of DPPC small unilamellar vesicles (SUVs), the alexa labeled free monomeric protein populations (fast component of diffusion model) decreases with concomitant increase in the membrane bound labeled protein that is the slow component. (g) The correlation functions of alexa

*Figure 2 continued on next page*

*Figure 2 continued*

488 maleimide labeled apo SOD1 in the absence (black) and presence of DPPC SUVs (red) where DPPC concentration was kept 500 nM. The green correlation curve corresponds to an intermediate DPPC concentration (100 nM). The inset shows the residual distributions of the correlation curves. (**h**) The hydrodynamic radii of free alexa 488-apo SOD1 and membrane bound labeled apo SOD1 were plotted against the concentrations of added DPPC SUVs. The average hydrodynamic radius of fast component that is free monomeric apo SOD1 ($r_{H1}$) was found to be 13.5 Å, whereas the average radius for slow membrane bound protein molecule ($r_{H2}$) was found to be 170 Å. The change of $r_{H1}$ and $r_{H2}$ with increasing DPPC SUV concentration remains invariant. (**i**) Percentage populations of membrane bound alexa-labeled protein variants were plotted against the concentrations of DPPC SUVs added to evaluate the binding affinities of the protein variants towards membranes. (**j**) Deconvoluted FTIR spectra of apo in membrane (DPPC SUV) bound condition. (**k**) Percentage of different secondary structural components in WT, apo, H121F, and H72F in the presence of DPPC SUVs are shown in this figure. n.s denotes nonsignificant change, while *** stands for significant change with p-value<0.001.

The online version of this article includes the following source data and figure supplement(s) for figure 2:

**Source data 1.** Structural characterization and membrane binding of SOD1 protein variants.
**Figure supplement 1.** Activity assay (pyrogallol auto-oxidation) for WT and other metal mutants of SOD1.
**Figure supplement 2.** Equilibrium unfolding transitions of H121F and H72F mutants using guanidinium hydrochloride as chemical denaturant.
**Figure supplement 3.** Acrylamide quenching experiments for the different variants.
**Figure supplement 4.** Far UV-CD spectra of the SOD1 variants.
**Figure supplement 5.** Second derivative of the FT-IR data obtained with WT SOD1.
**Figure supplement 6.** FTIR assessment of the H72F and H121F mutant.
**Figure supplement 7.** Membrane binding and conformational changes of the protein variants.

complex would behave inside the confocal volume. Using FCS we determined the correlation functions using 50 nM Alexa488Maleimide protein in the presence of increasing concentration of DPPC small unilamellar vesicles (SUVs) (*Figure 2g* showed the typical correlation functions of alexa labeled apo SOD1 in absence and presence of 100 nM and 500 nM DPPC SUVs). We fit the correlation functions using a two component diffusion model and the goodness of the fit was established using the randomness of the residual distribution. In this model, the fast and slow diffusing components corresponded to the free (with $r_{H1}$13.5 Å) and lipid bound protein ($r_{H2}$170 Å) respectively (*Figure 2h*). With increasing DPPC SUV concentration, the percentage of slow component increased (*Figure 2f*), which occurred at the expense of the fast component, and a sigmoidal fit of either of these components yield the values of $K_a$, which showed that the binding affinities followed the trend: apo $\geq$H72F>H121F>WT (*Figure 2i*, *Table 3*). Since FCS experiments required the use of labeled proteins in which the presence of bulky fluorescence dye can potentially influence the results, we complemented FCS binding data by measuring the tryptophan fluorescence of the SOD1 variants with increasing concentrations of DPPC SUVs. From the gradual enhancement of tryptophan fluorescence due to lipid binding, we calculated the binding affinities of all the protein variants towards membrane which showed comparable binding constants as obtained from our FCS experiments (*Figure 2—figure supplement 7b,c*). We then measured the Stern–Volmer constants using acrylamide quenching experiments of Trp32 fluorescence with protein variants in the absence ($K_{sv}$) and presence of ($K_{svm}$) membrane. The parameter $K_{sv}/K_{svm}$ was found maximum for the apo protein, and minimum for WT (*Figure 2—figure supplement 7d*, *Table 2*). H121F and H72F variants behaved like WT and apo protein, respectively. As observed by FT-IR, DPPC binding resulted in no or minimum change in conformation for WT and H121F proteins (*Figure 2—figure supplement 7e,f*), while a large decrease in beta sheet content with simultaneous rise in non-beta content, specifically alpha helical

**Table 2.** The values of Stern–Volmer quenching constants for Trp 32 residue of all the protein variants in the absence ($K_{sv}$,$M^{-1}$) and presence of DPPC SUVs ($K_{svm}$, $M^{-1}$) as obtained from acrylamide quenching for WT SOD1 and all the mutants including apo SOD1.

| Proteins | $K_{sv}$ | $K_{svm}$ | $K_{sv}/K_{svm}$ |
|---|---|---|---|
| WT SOD1 | 6.8 ± 0.1 | 5.7 ± 0.2 | 1.19 |
| H121F | 8.0 ± 0.1 | 6.3 ± 0.1 | 1.26 |
| H72F | 12.7 ± 0.3 | 7.0 ± 0.2 | 1.82 |
| apo SOD1 | 14.3 ± 0.1 | 7.6 ± 0.2 | 1.88 |

**Table 3.** Binding constants ($K_a$,$M^{-1}$) of the protein variants with model DPPC SUVs as obtained from the FCS study for WT SOD1 and all the mutants.

| Systems | Asssociation constants ($K_a$,$M^{-1}$) |
| --- | --- |
| WT SOD1 + DPPC SUV | $(4.1 \pm 0.1) \times 10^6$ |
| H121F + DPPC SUV | $(5.2 \pm 0.2) \times 10^6$ |
| H72F + DPPC SUV | $(9.6 \pm 0.4) \times 10^7$ |
| apo + DPPC SUV | $(9.8 \pm 0.1) \times 10^7$ |
| G37R-DPPC SUV | $(2.2 \pm 0.3) \times 10^6$ |
| I113T-DPPC SUV | $(8.8 \pm 0.2) \times 10^6$ |

content, was observed for the apo protein and H72F mutant (*Figure 2j,k Figure 2—figure supplement 7g*).

## Lipid vesicles accelerate aggregation kinetics of apo and zn-deficient mutants

Aggregation kinetics of WT, apo, and the mutant SOD1 in their TCEP reduced states were studied systematically both in the absence and in the presence of DPPC. A typical protein membrane ratio of 1:2 was maintained for all measurements involving membranes. For the initial assessment of the aggregation kinetics, the fluorescence intensity enhancement of amyloid marker Thioflavin T (ThT) was monitored. ThT is known to bind to protein aggregates with cross beta structure giving rise to a large increase in its fluorescence intensity. From the ThT fluorescence assay, we found that the WT protein does not aggregate, both in the absence or in the presence of membrane (*Figure 3a,b*). For the H121F variant in the absence of membrane, we found a small and slow enhancement of ThT fluorescence and the profile remained unchanged when we added the membrane (*Figure 3a,c*). In contrast, for apo and H72F variants, ThT assay showed large fluorescence increase and the kinetics followed typical sigmoidal patterns. The addition of membrane increased the rate of aggregation for both variants and a large decrease in the lag times. When compared between the apo and H72F variants, we found that the rate of aggregation is higher (i.e. with less lag time) for the apo protein (*Figure 3a,b,c Table 4*).

We then imaged using atomic force microscopy (AFM) the aggregates collected from the plateau regions of the aggregation kinetics (at a time point when the fluorescence of ThT was maximum [saturated] and did not change). Protein (P) aggregates will be denoted by $P_{agg}$, and $P_{aggm}$ to indicate if they are formed in the absence or presence of membranes respectively. For example, the aggregates of WT in the absence and presence of membranes would be denoted by $WT_{agg}$, and $WT_{aggm}$, respectively. AFM imaging also showed that in the absence of membrane, WT and H121F did not form aggregates, fibrillar, or otherwise (*Figure 3—figure supplement 1*), while large fibrillar aggregates were found to form with apo (*Figure 3d*) and H72F mutant (*Figure 3—figure supplement 1*). The average size of the fibrillar $apo_{agg}$ was found to be 1.8–2 µm with an average height of 20 nm. $H72F_{agg}$ showed similar morphology (*Figure 3—figure supplement 1*). Significant morphological differences were noticed for the aggregates of apo and H72F variants, in the absence (*Figure 3d,e*, *Figure 3—figure supplement 1*) and the presence of the membrane. The $apo_{aggm}$ appeared to exhibit network of thin aggregates (the average size was found to be 700–800 nm with an average height of 6–8 nm) which were found to be connected by the spherical DPPC vesicles (*Figure 3e*, inset; *Figure 3f*). To understand the effect of curvature on membrane binding and aggregation, we performed the binding experiments with H72F mutant using liposomes of different curvatures. We used DPPC SUVs (diameter ~78 nm), LUVs (diameter ~140 nm) and GUVs (~20 µm) for this study. Our results showed that with increasing curvature of the DPPC lipid vesicles, binding, and aggregation of H72F increased that is fibril formation rate and extents were found to be highest in the presence of SUVs and moderate for LUVs and lowest for GUVs (*Figure 3—figure supplement 2a,b*). To investigate the effect of H72F mutant toward curvature induction in GUVs, we studied the aggregation of H72F in the presence of GUVs and the final point aggregates were imaged through a transmission electron microscope. We found that as a result of the incubation with H72F, the size of the GUVs reduced many folds starting from an average size of 5–10 µm to about 1–2 µm.

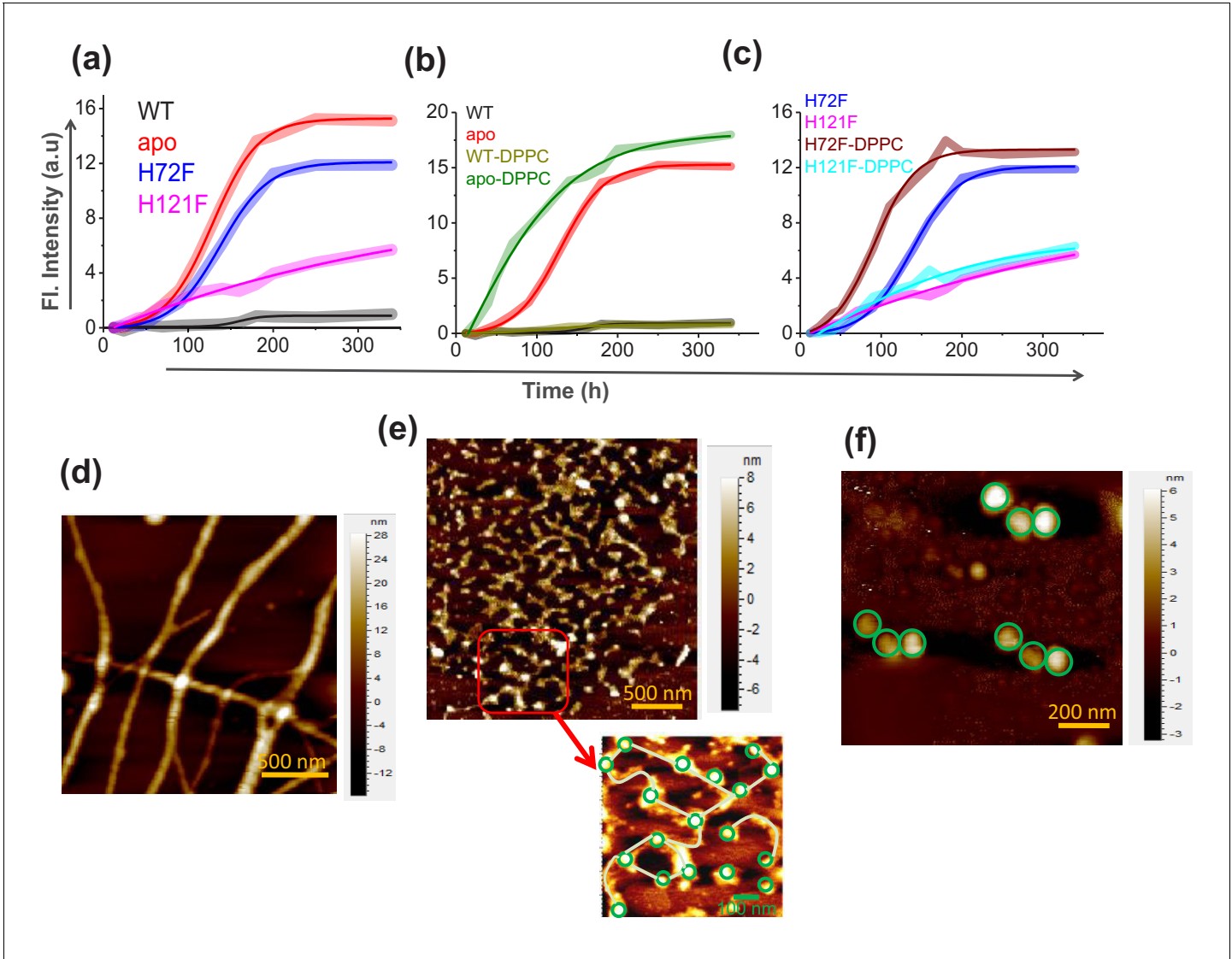

**Figure 3.** Aggregation of WT SOD1 and its mutants and membrane effects. (**a**) ThT fluorescence (at 484 nm) for the protein variants under reducing conditions to monitor the kinetics of aggregation. (**b**) Time-dependent increase in ThT fluorescence intensity of WT and apo both in the absence and in the presence of membrane (DPPC SUVs were used here as membrane). (**c**) Same as (**b**) but for H72F and H121F. Atomic force microscopy (AFM) images of the aggregates of apo SOD1 in the absence (**d**) and presence (**e**) of DPPC SUVs. These AFM images were taken at the plateau of the ThT aggregation curves. AFM images of apo$_{agg}$ showed linear fibrillar aggregates with an average size 1.8–2 μm. In contrast in the presence of membrane (apo$_{aggm}$), we found network of small fibrils, which were connected by DPPC vesicles (as drawn in the inset of **e**). It may be noted that the size and height (average diameter is 70 nm and average height is 7 nm) of the connecting spherical objects are similar to (**f**) AFM micrograph of the control DPPC SUVs, which showed distinct membrane structures with an average size of 70–90 nm.

The online version of this article includes the following source data and figure supplement(s) for figure 3:

**Source data 1.** Aggregation and effect of membrane curvature and composition on the aggregation behavior of SOD1 protein variants.
**Figure supplement 1.** AFM topographic images of the aggregates of different protein samples at the final points of aggregation.
**Figure supplement 2.** Effect of membrane curvature on binding and aggregation.
**Figure supplement 3.** Effect of membrane composition on binding and aggregation.

Numerous smaller vesicles are observed in the background that suggest a possible vesiculation of the GUV induced by H72F during the long incubation period (*Figure 3—figure supplement 2c,d*). Such vesiculation has also been reported earlier in some cases of other amyloid structures (*Meker et al., 2018*). These results suggested possible membrane remodeling induced by Zn-deficient H72F mutant, an aspect we would like to study in further detail. To determine the effect of

**Table 4.** Log-phase mid-points of different protein variants obtained from ThT assay.

| Systems | Log-phase mid-point (h) |
| --- | --- |
| WT SOD1 | Not detectable |
| WT SOD1 + DPPC SUV | Not detectable |
| H121F | Not detectable |
| H121F + DPPC SUV | Not detectable |
| H72F | 167.2 |
| H72F + DPPC SUV | 90.8 |
| apo SOD1 | 112.8 |
| apo + DPPC SUV | 55.9 |

membrane composition, we studied DPPG, which is a widely used negatively charged membrane with chain length identical to DPPC. Interestingly, our results showed that both binding and the rate of aggregation increased in the presence of a negatively charged membrane (*Figure 3—figure supplement 3*).

## Aggregates of apo and H72F variants that form in the presence of membrane show highest cellular toxicity and GUV deformation rates

We investigated the effect of the aggregates of different protein variants on cellular toxicity (by measuring cell viability) in general and on the cell membranes (using a number of spectroscopy and imaging assays) in particular. Cell viability was measured by the standard MTT assay. We used aggregates collected at the plateau region of the aggregation kinetics. MTT assay using SHSY5Y cell line showed minimum toxicity in terms of cell death for $WT_{agg}$, $WT_{aggm}$, $H121F_{agg}$, and $H121F_{aggm}$ (*Figure 4—figure supplement 1*). In contrast, $apo_{agg}$, and $H72F_{agg}$, showed significantly higher neuronal dead cell population which increased further with $apo_{aggm}$ and $H72F_{aggm}$, respectively. Although neuronal cell death is one of the decisive factors of aggregate toxicity, the severity of ALS has been found to depend on the extent of membrane perturbations, which may contribute to multiple events including (1) mitochondria-associated membrane (MAM) collapse and disruption (*Watanabe et al., 2016*), (2) the synaptic dysfunction due to impaired synaptic vesicles function toward neurotransmission (*Casas et al., 2016*; *Song, 2020*), and (3) the prion like spread of toxic aggregates between cells presumably through macropinocytosis (*Yerbury, 2016*; *McAlary et al., 2019*). Therefore, it is necessarily important to investigate in detail the protein aggregates induced membrane perturbation, which has never been addressed before. We used three different assays for the in vitro studies, viz. (1) phase-contrast microscopy using a membrane model of giant unilamellar vesicle (GUV) to probe how the presence of aggregates change their size and shapes, (2) a calcein release assay to probe aggregate-induced pore formation, and (3) FTIR to determine the molecular mechanism of the influence of different aggregates on the structure of lipids.

For the imaging assay, we used time based optical microscopic investigation to unveil how the addition of $P_{agg}$, and $P_{aggm}$ affects the size and shape of GUVs. GUV is widely used as a model membrane system, providing free-standing bilayers unaffected by support-induced artifacts yet with sufficiently low curvature to well mimic cellular membranes and mitochondrial membrane as well. An advantage of this assay comes from the use of phase contrast without requiring any external fluorophore label. We made the GUVs composed of DOPC:DOPE:PI:DOPS:CL in the ratio 4.5:2.5:1:0.5:1.5, which mimics mitochondrial membrane composition (*Gohil and Greenberg, 2009*). *Figure 4a* is a representative example of how apo-aggregates formed in the absence of membrane ($apo_{agg}$) behaved with the GUVs. In contrast, *Figure 4b* shows the influence of the apo-aggregates formed in the presence of membrane ($apo_{aggm}$) (*Figure 4—figure supplement 2*). *Figure 4c* shows an example of a control, in which WT samples ($WT_{agg}$, which should contain minimum aggregates population) were added to the GUVs. GUV images of *Figure 4a,b* clearly demonstrate that the aggregates attach on the surface of GUVs (shown by arrows) leading to membrane deformation and change in lamellarity. These two images also show the pore formation, which was further established by the contrast loss. A comparison between *Figure 4a,b* shows visually that membrane deformation

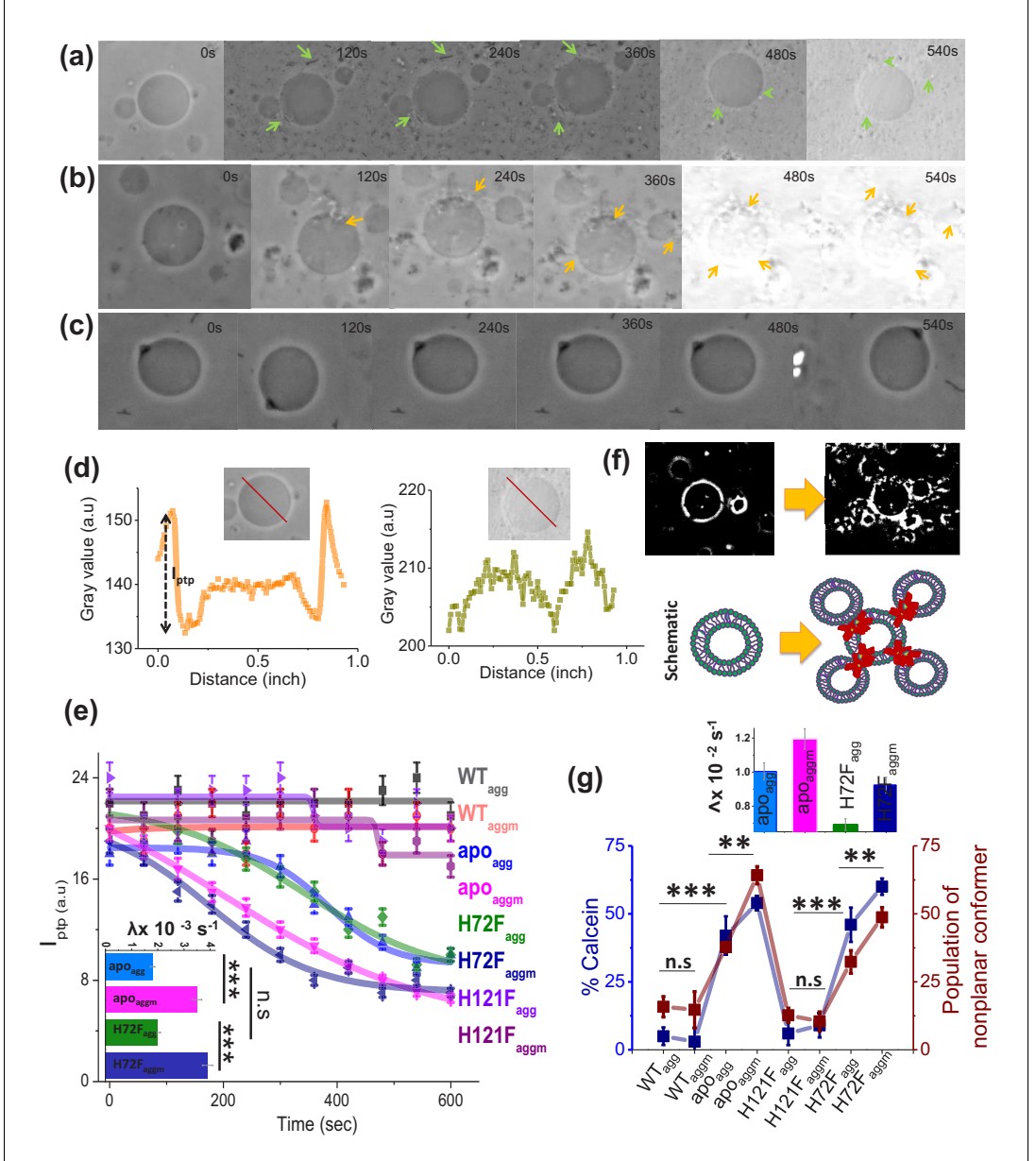

**Figure 4.** Membrane deformation by protein aggregates. Time variations of phase-contrast micrographs of a single GUV when GUVs were treated with (a) apo$_{agg}$, (b) apo$_{aggm}$, and (c) WT$_{agg}$. The images show gradual contrast loss, the loss of lamellarity and aggregate association with the vesicles for (a) and (b), while (c) does not show any change. (b) also shows few vesicles assemblies as induced by aggregates (also refer to a high contrast image below). The protein aggregates and their association with GUVs have been marked by green and yellow arrows. (d) The pictorial definition of I$_{ptp}$ and how I$_{ptp}$ changes for an intact (left) and porus (right) vesicle. (e) The values of I$_{ptp}$ are plotted for different protein aggregates with time. The inset of this figure shows the rate of deformation ($\lambda$, s$^{-1}$) for apo$_{agg}$/apo$_{aggm}$ and H72F$_{agg}$/H72 $_{Faggm}$. The typical sizes of the GUVs were ~30 µm. (f) High-contrast images of GUVs show vesicular assembly in the presence of apo$_{aggm}$, which is also schematically described in the figure below using a drawing. The schematic representation shows that apo$_{aggm}$ can act as a connector between multiple GUVs. (g) Plot of calcein leakage percentage and the population of nonplanar rotamers (these rotamers arise at 1367 cm$^{-1}$ vibrational frequency region of hydrocarbon lipid chains on treatment with different protein aggregates). Here, the subscripts agg and aggm stand for the aggregates of the respective protein species at the plateau region of the aggregation profiles, which were formed in the absence and presence of membrane (DPPC SUVs), respectively. The inset of this figure shows the rate of fluorescence growth($\wedge$, s$^{-1}$) due to aggregate induced pore formation mediated calcein dye leakage from the SUVs that mimic synaptic vesicles (composition of lipid: DOPC:DOPE:DOPS in the ratio 2:5:3). Here, n.s designates nonsignificant change, whereas ** stands for significant (p-value<0.01) and *** for highly significant (p-value<0.001). The error bars indicate the standard deviation of triplicate experimental data.

The online version of this article includes the following source data and figure supplement(s) for figure 4:

**Source data 1.** Toxicity of the aggregates .

*Figure 4 continued on next page*

*Figure 4 continued*

**Figure supplement 1.** MTT assay to detect the cell viability of neuronal cells (SHSY5Y) when treated with different aggregates of SOD1 protein variants.

**Figure supplement 2.** Time scale optical microscopic images of GUVs when these were treated with apo $_{aggm}$.

**Figure supplement 3.** High contrast images of the GUV when it was treated with apo$_{agg}$.

**Figure supplement 4.** Assembly of vesicles/GUV clustering when GUVs were treated with H72F$_{aggm}$.

**Figure supplement 5.** Change in I$_{ptp}$ of a single GUV in presence of DPPC SUVs (10 μM).

**Figure supplement 6.** Effect of aggregates on membrane deformation.

is more prominent (more damage) and faster (occurs at earlier time points) in the presence of P$_{aggm}$. All these changes in GUVs were found absent in *Figure 4c*, in which WT$_{agg}$, was used. For the quantification of the image data, we determined the difference in refractive indices between the exterior and interior of GUVs (represented by I$_{ptp}$, peak-to-peak intensity in *Figure 4d*; *Sannigrahi et al., 2019*). In the case of intact vesicles, the values of I$_{ptp}$ would be high (*Figure 4d*, left) due to the difference in sugar asymmetry between outside and inside of GUVs, which would disappear in the case of membrane deformation (*Figure 4d*, right). We plotted the time dependence of I$_{ptp}$ to determine quantitatively the membrane deformation kinetics by measuring the deformation rate constants, λ (*Figure 4e*, inset). The GUV deformation was insignificant and no detectable kinetics were found for WT$_{agg}$,, WT $_{aggm}$, H121F$_{agg}$, and H121F$_{aggm}$ (*Figure 4e*). On the other hand, apo$_{agg}$, and H72F$_{agg}$, exhibited significantly high deformation rate (λapo$_{agg}$ ~ $1.8 \times 10^{-3}$ s$^{-1}$ and λH72F$_{agg}$ ~ $2.1 \times 10^{-3}$ s$^{-1}$) and both kinetics appeared cooperative (sigmoidal behavior, *Figure 4e*). It is interesting to note that in the presence of apo$_{aggm}$ and H72F$_{aggm}$, the deformation rate increased significantly (λapo$_{aggm}$ ~ $3.5 \times 10^{-3}$ s$^{-1}$ and λH72F$_{aggm}$ ~ $3.9 \times 10^{-3}$ s$^{-1}$). More interestingly, both apo$_{agg}$ and apo$_{aggm}$ showed a tendency to create attachments between vesicles to generate vesicular assembly and co-operative deformations (high-contrast images in *Figure 4f*, which is also shown by a schematic drawing). The image analysis and visualization of the GUVs in presence of apo$_{aggm}$ suggested that the co-operative vesicular clustering and deformation presumably occurred through allosteric communications mechanism by the aggregates (*Figure 4—figure supplement 2*). We found that apo$_{aggm}$ and H72F$_{aggm}$ are more efficient towards inducing vesicular assembly and deformations (*Figure 4f*, *Figure 4—figure supplement 3*, *Figure 4—figure supplement 4*). To rule out the effect of only SUVs (which are present along with the protein), we performed a control experiment by treating the GUVs with similar concentration of DPPC SUVs as employed for the formation of P$_{aggm}$. Our results showed insignificant changes in I$_{ptp}$ values in the presence of DPPC SUVs (*Figure 4—figure supplement 5*), further highlighting that P$_{agg}$ and P$_{aggm}$ are responsible for the GUV perturbations.

Electron microscopic images of the synapses infused with ALS variants, like G85R, showed vacant active zones (AZs) and occasional abnormal membranous structures, whereas there occurred no reduction in the synaptic vesicles number in case of WT SOD1 (*Song, 2020*). Similar observation was found previously by *Wang et al., 2009*. Using *C. elegans* as model system, they showed that the neuronal toxicity in ALS appears due to synaptic dysfunction that occurs because of misfolded and aggregated disease mutant driven lowering in the number of organelles including synaptic vesicles and mitochondria. The pore formation and vesicle rupture by aggregates may be a possible reason for the reduced synaptic vesicle population in case of ALS disease mutants. To investigate this issue, we prepared calcein entrapped SUVs composed of DOPE, DOPS, and DOPC at the molar ratio 5:3:2 to mimic the synaptic vesicle composition and curvature (*Fusco et al., 2016*). In this assay, we measured the percentage of calcein leakage from the dye entrapped inside lipid vesicles (*Figure 4g*, *Figure 4—figure supplement 6*). The extent of calcein leakage after the treatment of different protein aggregates followed the trend similar to what was observed by GUV micrographic observation (*Figure 4e*), which suggested that the membrane rupture is facilitated in significantly higher rate and extent by apo$_{aggm}$ and H72F$_{aggm}$, while compared with apo$_{agg}$ and H72F$_{agg}$, respectively (*Figure 4g*, inset, *Figure 4—figure supplement 6a*).

In order to induce membrane deformation as observed by previous two assays, the aggregates need to inflict substantial changes in lipid structure. To determine the extent of conformational change the lipid molecules experience by protein variants, we used ATR-FTIR to measure quantitatively the populations of different rotamers in a general planar trans-oriented phospholipid bilayer of

DPPC. CH$_2$ wagging band frequency (1280–1460 cm$^{-1}$) of the hydrocarbon tail region of the bilayer was carefully monitored for this purpose (*Lewis and McElhaney, 2013*; *Maroncelli et al., 1982*). The results show a significant increase in the populations of nonplanar kink+gtg$'$ rotamers (this band appears at 1367 cm$^{-1}$) when planar lipid bilayer was treated with apo$_{agg}$/apo$_{aggm}$ and H72F$_{agg}$/H72F$_{aggm}$ (*Figure 4g*; *Figure 4—figure supplement 6b,c*). We also found that apo$_{aggm}$/H72F$_{aggm}$ exerted greater effect on lipids than apo$_{agg}$/H72F$_{agg}$. Interestingly, differences in the populations of non-planar conformers are found to be similar to the variation in the extent of calcein release induced by protein variants (*Figure 4g*). This observation provides preliminary evidence that both events occur by similar trigger (presumably the membrane attachment by the aggregates).

## Aggregation kinetics and aggregate-induced toxicity studies of ALS disease mutants

In the previous few sections using WT, apo, H121F, and H72F, we established that the WT (containing both Zn and Cu) and the Zn bound H121F protein did not aggregate or induced toxicity through membrane deformation. In contrast, the removal of either Zn (H72F mutant) or both metals (the apo protein) resulted in strong membrane association, aggregation and induction of cellular toxicity. The data clearly suggest that Cu plays secondary roles in aggregation induced toxicity, while Zn pocket destabilization acts in concert with membrane-induced conformational change resulting in aggregation and toxic gain of function. Since these experiments validate successfully the predictions from the statistical mechanical model, we subsequently wanted to understand this behavior could be generalized in ALS disease mutants.

For this purpose, we used both in silico and experimental approaches. For the in silico method, we selected fifteen disease mutants of ALS, whose structures are available in the protein data bank (PDB) (*Table 5*). From the crystal structures, we determined the distance between the mutation site and Zn (and Cu) for all these disease mutants. In addition, we calculated the transfer free energies, that is the theoretical membrane association energies ($\Delta G_{Tr}$), of these mutants using the OPM server. We found that the negative values of $\Delta G_{Tr}$ decreased linearly with the increased distance of Zn site for these mutants, while it remained non-variant with the distance of Cu site (*Figure 5a*). We also found that the severity of ALS mutants (defined here as the survival time in year of ALS patients after

**Table 5.** SOD1 disease mutants, their corresponding distance parameters in terms of the distances of mutational stress points from the Zn and Cu center, and the membrane binding energies of the disease mutants.

| Mutants | Mutational points distances from zn center (Å) | Mutational points distances from Cu Center (Å) | Membrane binding energies (ΔG, kcal/mole) |
|---|---|---|---|
| A4V | 19.6 | 19.3 | −1.6 |
| C6A | 17 | 15.7 | −2.1 |
| G37R | 23.7 | 17.2 | −1.0 |
| L38V | 22.2 | 13.6 | −1.2 |
| H43R | 17.8 | 9.9 | −1.0 |
| H46R | 7.8 | 10.2 | −2.3 |
| H80R | 4.2 | 10.3 | −2.3 |
| G85R | 8.6 | 8.4 | −2.0 |
| G93A | 20.5 | 22.7 | −1.8 |
| C111A | 11.9 | 18.3 | −3.3 |
| C112S | 19.7 | 17.6 | −1.3 |
| I113T | 16 | 18.5 | −3.2 |
| D124V | 9 | 13 | −2.1 |
| H72F | 2.9 | 8.2 | −3.1 |
| H121F | 14.4 | 5.6 | −1.9 |
| S134N | 9.6 | 10.9 | −1.6 |
| C147S | 16.3 | 10.3 | −1.7 |

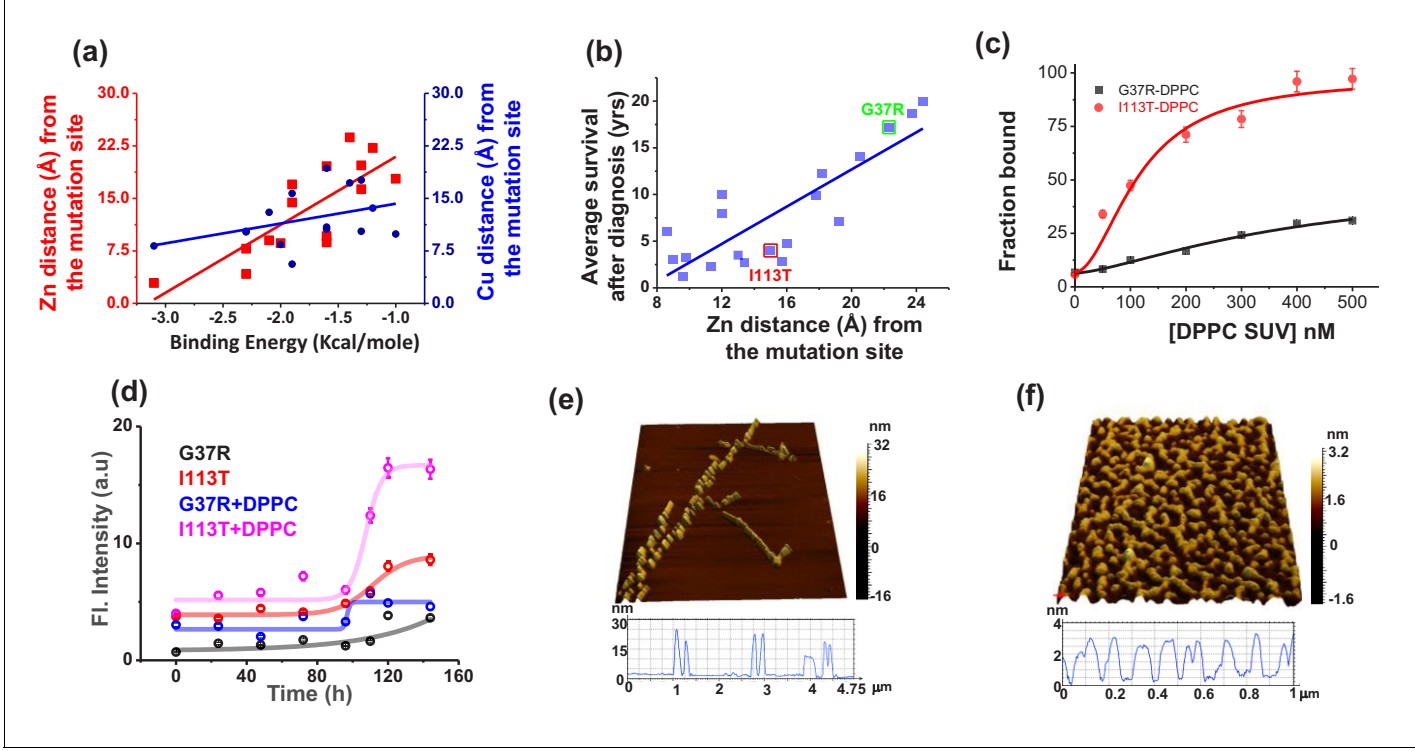

**Figure 5.** Validation in ALS disease mutants. (**a**) Computational validation. The distances of the disease mutation sites from the Zn (red) and Cu (blue) co-factor were plotted against the binding energy towards membrane, which were computed. The binding energy decreased linearly as the distance between Zn and mutation site increased (red curve), while there was no significant change for Cu (blue curve). The distance information for these mutants was calculated from their solved structures. (**b**) The ALS disease severity in terms of average survival time after diagnosis has been plotted against the distance parameters of the mutation points from Zn centre for the disease mutants.(**c**) percentage populations of membrane bound alexa-labeled protein variants which were obtained from the two component diffusion model fitting of the FCS data for G37R and I113T were plotted against the concentrations of DPPC SUVs added to evaluate the binding affinities of the protein variants towards membranes. (**d**) Fibril formation kinetics. ThT fluorescence spectra of G37R and I113T were plotted in the absence and presence of DPPC SUVs. The 3D AFM images of I113T in absence (**e**) and presence (**f**) of membranes. Error bar indicates the standard deviation for the triplicate experiments.

The online version of this article includes the following source data and figure supplement(s) for figure 5:

**Source data 1.** Conformational characterization, membrane binding, aggregation and toxicity of ALS disease mutants.
**Figure supplement 1.** Conformational changes in disease mutants.
**Figure supplement 2.** Effect of point mutations (here disease mutations) of the proteins flexibility, conformational stability and dynamics as predicted using DynaMut webserver (http://biosig.unimelb.edu.au/dynamut/).
**Figure supplement 3.** Aggregate morphology of G37R mutant.
**Figure supplement 4.** MTT assay to detect the cell viability of neuronal cells (SHSY5Y) when treated with different aggregates of SOD1 disease mutants.
**Figure supplement 5.** Effect of disease mutant aggregates on membrane deformation.
**Figure supplement 6.** Comparative study of membrane binding and aggregation of different disease mutants.

diagnosis) correlates nicely with the distance of the Zn sites (*Figure 5b*; *Wang et al., 2008*). For the experimental validation, we used two representative ALS disease mutants of varying distance between the mutation site and Zn. The mutations, namely G37R (mutation site-Zn distance 24 Å, less severe) and I113T (mutation site-Zn distance 16 Å, more severe) (*Figure 5b*) are well studied in literature (*Milardi et al., 2010*; *Krishnan et al., 2006*; *Banci et al., 2009*). Using FCS we also measured the $K_a$ values of these two disease mutants which showed greater binding affinity for I113T (*Figure 5c*, *Table 3*). Structural characterization using FTIR showed greater extent of alpha helical content in I113T compared to G37R. On the other hand, further increase in the alpha helical structure in I113T was observed on interaction with model membrane. In contrast, we did not observe any membrane induced structural change for G37R (*Figure 5—figure supplement 1*). To understand the internal structural changes due to the disease mutations, we performed some in silico study using DynaMut webserver (http://biosig.unimelb.edu.au/dynamut/) to predict the effects of

mutational stress on the conformational dynamics, stability and flexibility of protein (***Rodrigues et al., 2018***). The results suggested that G37R mutation increased the rigidity of different regions (6–16, 30–42, 80–86 aa) of the protein, whereas significant increase in flexibility of the loop IV and VII region (48–52, 130–150) was observed in case of I113T mutation (***Figure 5—figure supplement 2***). The predicted changes in folding free energy ($\Delta\Delta G$, kcal/mol) for I113T was found to be much more negative than G37R ($\Delta\Delta G(\Delta G_{WT} - \Delta G_{mutant})$) suggesting that mutation at 113th position destabilized the protein more in comparison to 37$^{th}$ position (***Figure 5—figure supplement 2***). The vibrational entropy change ($\Delta\Delta S$) in I113T was found to be +0.188 kcal mol$^{-1}$ K$^{-1}$ while −0.404 kcal mol$^{-1}$ K$^{-1}$ was found for G37R indicating the increase in flexibility for I113T and decrease in flexibility for G37R (***Figure 5—figure supplement 2***). Thus the increase in alpha helical content in I113T mutant arises due to the increase in flexibility in the membrane interacting loop regions as suggested by the DynaMut server. Subsequently, we used ThT fluorescence to probe the aggregation kinetics of G37R and I113T in the absence and presence of DPPC SUVs. Results in the absence of membrane showed significantly higher aggregation for I113T when compared to G37R. A notable increase in the rate and extent of aggregation was observed for I113T under membrane environment, whereas for G37R, the change was not significant (***Figure 5d***). AFM imaging showed linear fibrillar aggregate for I113T in the absence of membrane (average length is ~1.5 μm and height is 26 nm), whereas fibrilar network was observed when aggregation occurred in the presence of DPPC SUVs (average height is 3.2 nm) (***Figure 5e,f***). In contrast, small nonfibrilar aggregates were found for G37R mutant both in the absence and in the presence of membranes (***Figure 5—figure supplement 3***). Finally, we studied the toxic effects of the aggregates of G37R and I113T, which formed in the absence and presence of membrane. Using MTT, we showed that I113T aggregates in the absence of membrane were less toxic than I113T aggregates in the presence of membrane (***Figure 5—figure supplement 4***). G37R exhibited minimal toxicity both in the absence or presence of membrane (***Figure 5—figure supplement 4***). We observed for these mutants a nice correlation between the extent of calcein release and the populations of nonplanar kink+gtg$'$ conformers (***Figure 5—figure supplement 5***).

For further validation, we studied the membrane association and aggregation of a third disease mutant G85R. Earlier reports suggested that the survival time for G85R (6 years) remains between G37R (17 years) and I113T (4.3 years) (***Wang et al., 2008***). Our computational and experimental binding study using FCS (***Figure 5—figure supplement 6***) showed that the binding affinity of G85R (−2 kcal mol$^{-1}$) is intermediate between that of G37R (−1 kcal mol$^{-1}$) and I113T (−3.2 kcal mol$^{-1}$). In addition, AFM imaging (***Figure 5—figure supplement 6***) suggests that G85R aggregation behavior in membrane environment remains between mild G37R and severe I113T mutants.

## Discussion

### The cofactor-derived membrane association model

In this work, we effectively bring a large number of studies together in a coherent framework to address the different roles of two metal cofactors (Cu and Zn) and membrane binding on the aggregation and induced toxicity of SOD1. Based on steady-state fluorescence, acrylamide quenching, and FT-IR data, the investigated proteins could be classified as (1) WT and WT-like mutant, H121F and (2) apo and apo like mutant, H72F. The battery of experimental approaches clearly validates the prediction from the bWSME model that Zn is largely responsible for the conformational stability of SOD1. The insertion of Zn at the loop IV region stabilizes the loop while reducing the aggregation propensity of the apo protein. We believe that the stabilization of this long loop may play an important role in SOD1 aggregation biology and its relevance in ALS. It may be noted that shortening of the loop has been shown to increase the stability of the protein (***Yang et al., 2018***). This is also important to point out that SOD1 variants from thermophiles (like SOD5 from *C. albicans* SOD5) have shorter electrostatic loops compared to human SOD1 (***Gleason et al., 2014***).

In contrast, the absence of Cu does not seem to have any significant effect on the stability of the protein, and the Zn containing Cu-deficient protein behaves like WT. We derive a possible maturation/aggregation landscape of SOD1 in the presence of membrane employing data from the current work and available literature (***Figure 6***). Since all experiments presented here were carried out using the reduced protein, this scheme does not take dimerization into consideration. SOD1 in its metal

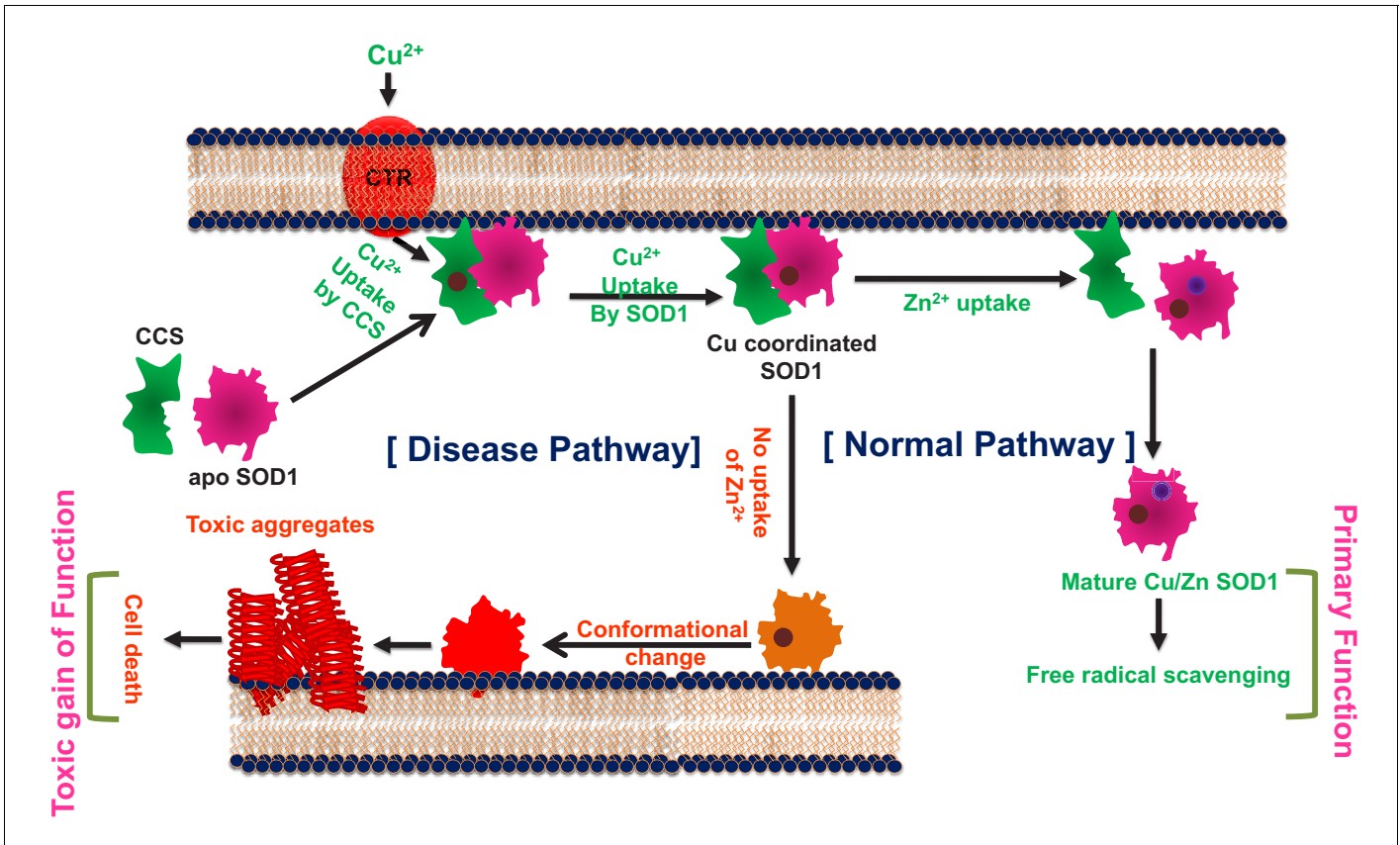

**Figure 6.** The cofactor-derived membrane association model of SOD1 primary and gain of function.

free state (apo protein) has been shown to be flexible and inherently dynamic. The sequence sites (48–80 and 120–140) of low folding probability predicted by bWSME calculation (*Figure 1b*) have significant overlap with the sequence sites (45–70 and 125–142) of high membrane binding as calculated by OPM (*Figure 2e*). It has been found that truncation at Leu126 of C-terminus resulted in a protein with several transmembrane helices (*Lim et al., 2015*). We found that the apo protein has high membrane binding possibility which is directly supported by membrane binding data (*Figure 2i*, *Figure 2—figure supplment 7b,c*). Membrane bound apoSOD1 with an optimized orientation is a requisite for the Cu insertion to take place. Since biological systems have very-low-free Cu salt (*Rae et al., 1999*), Cu coordination to SOD1 occurs through Cu chaperone protein (CCS) which transfers Cu to apo SOD1 using membrane as a scaffold (*Pope et al., 2013*). Membrane scaffolding decreases the directionality of metal transport compared to a three-dimensional search of the metal ions. The domain I of CCS binds Cu through two cysteine residues. Since Cu binding affinity of CCS is less than that of SOD1 (which recruits three or four cysteine residues), Cu transport occurs from CCS to SOD1, and not the other way (*Banci et al., 2010*). Zn binding, which occurs in the next step, stabilizes the loop regions. This event has a few important consequences. First, it has been shown that both CCS and SOD1 exhibit binding to Zn (*Proescher, 2008*). While the absence of Zn favors heterodimer formation with SOD1 (CCS-SOD1), the presence of Zn facilitates the homodimer (CCS-CCS) configuration. Second, as shown in this paper, Zn coordinated protein has little or minimum membrane binding and hence Zn bound SOD1 is removed from bi-layer. The third consequence comes from the reduced Zn affinity (which may come from a Zn compromised mutation or other reason for the sporadic forms of the disease), which would result in a misfolded membrane bound protein that is aggregation prone and potentially toxic.

As evident, the proposed scheme is somewhat Zn centric in which Cu coordination plays minimum role in the aggregation centric disease process. Partial support of this comes directly from the presented data of the distance dependence between ALS mutations sites and Cu. Also, the mutant

proteins without Cu (but with Zn site coordinated, e.g. H121F or G37R) do not show aggregation, neither they induce toxicity as studied by our assay systems. Earlier works have also shown that Cu binding has no effect on SOD1 folding (*Bruns and Kopito, 2007*). More importantly, it has been shown that CCS knock-out does not affect the onset of the disease or the life span in SOD1 transgenic mice (*Subramaniam et al., 2002*).

It may be important to consider that, although more than 140 SOD1 mutants are known in ALS, the disease is predominantly sporadic. Nevertheless, the disease initiation may still be done by a metal free apo (or Zn free) configuration, which can be generated from the WT protein as a result of a different trigger (and not by a genetic factor). The results obtained from two ALS mutants (I113T and G37R) can be discussed in this context. The present data show that I113T is similar to a zinc deficient protein, with high affinity toward membrane and higher aggregation propensity. In contrast, G37R behaves more like a WT protein. A comparison between the disease phenotype show that I113T is the second most common ALS mutant with average survival of 4.3 years, while for G37R, the average survival increases to 17 years. These results are in excellent correlation with the presented scheme.

Another interesting correlation of the presented scheme comes from a result that Zn supplement with a moderate dose of 60 mg kg$^{-1}$ day$^{-1}$ can increase the days of survival in a transgenic mouse experiment (*Ermilova et al., 2005*). Another approach could be using small molecules which would compete with the metal free protein towards membrane binding, an approach used recently in a Parkinson's disease model (*Fonseca-Ornelas et al., 2014*). While any drug development initiative targeting ALS and other neurodegenerative diseases suffer from the complications of diseases heterogeneity, poorly understood molecular mechanism and complex delivery avenues, a collaborative and integrative effort involving science and clinic, may be needed to find a successful solution.

## The membrane connection

The molecular mechanism associated with ALS induced toxicity has been extensively studied and widely debated (*Broom, 2012*). SOD1 aggregates have been found in ALS patient samples (*Gruzman et al., 2007*; *Guareschi et al., 2012*). Mutant SOD1 aggregates have been shown to transfer from cell to cell using a prion like propagation mechanism (*Münch et al., 2011*). It has also been shown that the aggregation induced toxicity of SOD1 variants can occur through its attachment on mitochondrial membrane surface in transgenic ALS mice (*Zhai et al., 2009*). In parallel, pore formation at the membrane and abnormal ion mobility have been found with SOD1 aggregation (*Ray et al., 2004*). Although the above reports directly link a membrane connection with SOD1 aggregation (and presumably with ALS)-unlike other neurodegenerative diseases like AD and PD, membrane-induced aggregation studies with SOD1 have been limited (*Choi et al., 2011*).

Shahmoradian and colleagues reported that the structure of Lewy bodies in Parkinson's disease consists of α-synuclein and lipid vesicle clusters instead of the long-assumed amyloid fibril core (*Shahmoradian et al., 2019*). Using correlative light and electron microscopy (CLEM), they show that the vast majority of Lewy bodies actually consist of clusters of various membranous compartments, instead of amyloid fibrils as previously assumed. Assembly of synaptic vesicles has been shown recently by *Fusco et al., 2016* in which two different regions of a protein molecule can be used as a 'double anchor' to induce the assembly. We think three particular finding can be discussed in this connection. First, the presented membrane deformation assay using GUV clearly shows the formation of vesicles assemblies (*Figure 4f*) induced by apo$_{aggm}$ and H72F$_{aggm}$. Second, for both protein variants, relatively small aggregates of the proteins (formed in the presence of the membrane) and not the large fibrilar network (formed in the absence of membrane), favored vesicles assemblies. Third, the smaller sized aggregates formed in the presence of membrane seemed to induce more toxicity than the fibrils (formed in the absence). From the above three considerations, it is easy to envision that the anchoring ability of the aggregates toward vesicles assembly would be more efficient for a small-sized aggregates when compared to a fibril. There is an increased interest in the role of the number and nature of membrane lipids in shaping the aggregation landscape of different proteins in neurodegenerative diseases. Given the prion-like propagation in these diseases, our work underscores how characterization of protein–lipid interactions could enhance our molecular understanding of cellular toxicity and enable the identification of therapeutic molecules to mitigate the damage.

# Materials and methods

## Key resources table

| Reagent type (species) or resource | Designation | Source or reference | Identifiers | Additional information |
|---|---|---|---|---|
| Cell line (*Homo sapiens*) | SH-SY5Y | National Centre for Cell Science, Pune, India | https://www.nccs.res.in/index.php/Nationalrepos/Repocellines | Neuroblastoma cell line |
| Recombinant DNA reagent | H72F (plasmid) | This paper | | Clone used to generate H72F mutant (page 21) |
| Recombinant DNA reagent | H121F (plasmid) | This paper | | Clone used to generate H72F mutant (page 21) |
| Recombinant DNA reagent | I113T (plasmid) | Genscript | | |
| Recombinant DNA reagent | G37R (plasmid) | Genscript | | |
| Recombinant DNA reagent | G85R (plasmid) | Genscript | | |
| Sequence-based reagent | H72F_F | Biotech Desk | PCR primer | cctcactttaatcctctatccagaaaattcggtgggccaaagg |
| Sequence-based reagent | H72F_R | Biotech Desk | PCR primer | cctttggcccaccgaattttctggatagaggattaaagtgagg |
| Sequence-based reagent | H121F_F | Biotech Desk | PCR primer | ggccgcacactggtggtctttgaaaaagcagatgactt |
| Sequence-based reagent | H121F_R | Biotech Desk | PCR primer | aagtcatctgcttttttcaaagaccaccagtgtgcggcc |
| Commercial assay or kit | QuickChange XL site directed mutagenesis kit | Agilent | Cat# 200516 | |
| Commercial assay or kit | Vybrant MTT cell proliferation assay kit | Invitrogen | Cat# V-13154 | |
| Chemical compound, drug | 1,2-Dipalmitoyl-sn-glycero-3-phosphocholine (DPPC) | Avanti Polar Lipids Inc. | Cat# 850355 | |
| Chemical compound, drug | 1,2-Dioleoyl-sn-glycero-3-phosphoethanolamine (DOPE) | Avanti Polar Lipids Inc. | Cat# 850725 | |
| Chemical compound, drug | 1,2-Dioleoyl-sn-glycero-3-phospho-L-serine (DOPS) | Avanti Polar Lipids Inc. | Cat# 840035 | |
| Chemical compound, drug | Cardiolipin | Avanti Polar Lipids Inc. | Cat# 710335 | |
| Chemical compound, drug | 1,1'-Dioctadecyl-3,3,3',3'-tetramethylindotricarbocyanine iodide (DiIC-18(3)) | Invitrogen | Cat# D3911 | |
| Chemical compound, drug | Isopropyl ß-D-1-thiogalactopyranoside (IPTG) | Sigma Aldrich | Cat# I6758 | |
| Chemical compound, drug | Thioflavin T | Merck | Cat# T3516 | |
| Chemical compound, drug | Calcein AM | Merck | Cat# 17783 | |
| Software, algorithm | OPM server | Orientations of Proteins in Membranes | https://opm.phar.umich.edu/ppm_server | |
| Software, algorithm | AGGRESCAN | AGGRESCAN | http://bioinf.uab.es/aggrescan/ | |
| Software, algorithm | Image J | National Institutes of Health | https://imagej.nih.gov/ij/ | |
| Other | Alexa Fluor 488 C5 maleimide | Invitrogen | Cat# A10254 | |

## Reagents

1,2-Dipalmitoyl-sn-glycero-3-phosphocholine (DPPC), 1,2-dioleoyl-sn-glycero-3-phosphoethanol-amine(DOPE), phosphoinositol (PI),1,2-dioleoyl-sn-glycero-3-phospho-L-serine (DOPS), and cardiolipin (CL) were purchased from Avanti Polar Lipids Inc (Alabaster, AL). 1,1′-Dioctadecyl-3,3,3′,3′-tetramethylindotricarbocyanine iodide (DiIC-18(3)) was purchased from Invitrogen (Eugene, OR). All other necessary chemicals were obtained from Aldrich (St. Louis, MO) and Merck (Mumbai, India).

## Experimental methods

### Expression and purification of SOD1

Recombinant SOD1 was over-expressed in *E. coli* (BL21 DE3 strain). The over-expression of SOD1 was induced with 1 mM isopropyl-1-thio-$\beta$-D-galactopyranoside (IPTG). The induction was coupled with metalation where 1 mM CuSO$_4$ was added directly to the Luria-Bertani culture media so as to ensure proper metal loading over the protein. Followed by induction, the cells were allowed to grow for 3.5 hr. The cells were pelleted down by centrifuging at 6000 rpm for 15 min at 4°C followed by re-suspension in pre-chilled lysis buffer (20 mM Tris-HCl +500 mM NaCl, pH 8.0). After thorough re-suspension in lysis buffer, the cells were subjected to sonication (20 pulses, each of 30 s pulse time and an interim time frame of 1 min). Unbroken cells and debris were removed by another act of centrifugation at 10,000 rpm for 10 min. The soluble fraction obtained thereafter was carefully removed and allowed to bind to Ni-NTA agarose resin. The Ni-NTA column was washed with 40 mL wash buffer (20 mM Tris–HCl, 500 mM NaCl and 50 mM imidazole, pH 8.0) followed by elution with 20 mM Tris HCl, 500 mM NaCl, and 500 mM imidazole, pH 8.0. The eluted fractions were pooled according to their tentative protein content as per their absorbance at 280 nm. The post-elution fractions were subjected to dialysis in 20 mM Na-phosphate buffer pH 7.5. In all our protein concentration measurements ultraviolet spectroscopy was deployed and SOD1 concentration was determined by considering the monomeric molar extinction coefficient of 5500 M$^{-1}$ cm$^{-1}$ at 280 nm (*Wright et al., 2013*). The identity of the protein was confirmed by SDS–PAGE and MS/MS. The metal content was confirmed by the atomic absorption spectroscopy and activity assays as reported earlier (*Marklund and Marklund, 1974*).

### Site-directed mutagenesis

The recombinant plasmid pET-19b, containing the gene for hSOD1 with a poly Histidine tag at the N-terminal end, was used as a template for mutagenesis using the QuickChange XL Site-Directed Mutagenesis Kit (Stratagene, Agilent). The mutagenic primers containing the mutations (shown in bold type) for the replacement of His-72 with a phenylalanine residue (H72F) and His-121 with a phenylalanine residue (H121F) are indicated below. Both primers were annealed to the same target sequence on opposite strands of pET-19b. The site-directed mutagenesis was performed using protocol as described by the manufacturer. The clone used for production of the H72F and H121F mutant was confirmed by DNA sequencing. Proteins were expressed in BL21 (DE3) pLysS *E. coli* cells by induction with 0.5 mM IPTG at 37°C for 4 hr. Cells were resuspended in ice cold 20 mM Tris-HCl, 500 mM NaCl, pH 8.0, containing protease inhibitor (2 mM PMSF) and lysed by sonication. Unbroken cells and debris were removed by centrifugation at 10000 g for 10 min. Binding of soluble proteins from the supernatant to Ni-NTA agarose resin (Qiagen) was done overnight. The Ni-NTA flow through was collected for analysis. The Ni-NTA resin was washed with 40 mL of wash buffer (20 mM Tris–HCl pH 8, 500 mM NaCl, 50 mM imidazole). Elution was done with wash buffer containing 500 mM imidazole. The designed primers for different mutants are as follows:

> H72F 5′-cctcactttaatcctctatccagaaaattcggtgggccaaagg-3′
> H72F_antisense 5′-cctttggcccaccgaattttctggatagaggattaaagtgagg-3′
> H121F 5′-ggccgcacactggtggtctttgaaaaagcagatgactt-3′
> H121F_antisense 5′-aagtcatctgctttttcaaagaccaccagtgtgcggcc-3′

### Preparation of apo SOD1

Preparation of apo enzyme from WT SOD1 by metal chelation was carried out following the earlier reported protocol (*McCord and Fridovich, 1969*). Overnight dialysis of WT SOD1 in 50 mM Na-

acetate, 10 mM EDTA, pH 3.8 was done so as to ensure proper removal of metal ions. EDTA was removed by repeated dialysis in 50 mM Na-acetate, pH 5.2 and in 20 mM Na-phosphate, pH 7.5. The de-metalation was ensured by an activity assay of SOD1 indexing photo-oxidation of pyrogallol at pH 8.

## Enzyme assay for SOD1

Inhibition of superoxide anion ($O_2^-$) mediated pyrogallol auto-oxidation in alkaline buffer (Tris–cacodylate, pH 8.2) by SOD1 was performed for measuring the activity of the metalloenzyme (*Marklund and Marklund, 1974*). Absorption of the oxidized product at 420 nm was monitored with respect to time to assess the activity of Cu/Zn SOD1. Extent of pyrogallol auto-oxidation was measured from the ratio of product absorption at 420 nm in the absence and presence of SOD1. Time plot of 0.2 mM pyrogallol was constructed. Then protein (100 nM) was added and time plot was recorded. Product formation in absence of SOD1 was taken as the reference value of 100.

## Preparation of SUVs

The appropriate amount of lipids in chloroform (concentration of stock solution is 25 mg mL$^{-1}$) was transferred to a 10 mL glass bottle. The organic solvent was removed by gently passing dry nitrogen gas. The sample was then placed in a desiccator connected to a vacuum pump for a couple of hours to remove traces of the leftover solvent. A required volume of 20 mM sodium phosphate buffer at pH 7.4 was added to the dried lipid film so that the final desired concentration (10 mM) was obtained. The lipid film with the buffer was kept overnight at 4°C to ensure efficient hydration of the phospholipid heads. Vortexing of the hydrated lipid film for about 30 min produced multilamellar vesicles (MLVs). Long time vortexing was occasionally required to make uniform lipid mixtures. This MLV was used for optical clearance assay and dynamic light scattering (DLS) experiment. For preparing the SUV, as-formed MLV was sonicated using a probe sonicator at an amplitude 45% for 30 min, and after that, sample was centrifuged at 5000 rpm to sediment the tungsten artifacts, and finally, it was filtered by 0.22 μm filter unit. Size of the vesicles was measured by DLS to be ~70 nm in diameter.

## Tryptophan-quenching experiments

Steady-state fluorescence spectroscopy and acrylamide quenching measurements in free and in membrane bound conditions were carried out using a PTI fluorimeter (Photon Technology International, USA). A cuvette with 1 cm path length was used for the fluorescence measurements. For the tryptophan fluorescence quenching experiments, an excitation wavelength of 295 nm was used to eliminate the contributions from tyrosine fluorescence. Fluorescence data were recorded using a step size of 1 nm and an integration time of 1 s. Excitation and emission slits were kept at 5 nm in each case. Emission spectra between 305 nm and 450 nm were recorded in triplicate for each experiment. Typical protein concentration of 10 μM was used for each quenching experiment and 1:200 protein–lipid molar ratio was maintained. The protein solutions were incubated at room temperature for 1 hr and then titrated using a stock of 10M acrylamide. Necessary background corrections and inner filter effect corrections were made for each experiment.

## Acrylamide-quenching data analysis

Assuming $I$ and $I_o$ are the tryptophan fluorescence intensity of the proteins in the presence and absence of acrylamide concentration $[Q]$, the Stern–Volmer equation (*Lakowicz, 1999*) can be represented as follows:

$$\frac{I_0}{I} = 1 + K_{SV}[Q] \tag{1}$$

where $K_{SV}$, is the Stern–Volmer constant, which can be determined from the slope of the linear plot of $I_0/I$ versus acrylamide concentrations $[Q]$.

## Fourier transform infrared spectroscopy

Fourier transform infrared spectroscopy (FTIR) spectra of WT SOD1, apo SOD1, and mutants in the absence and presence of denaturants were acquired using a Bruker 600 series FTIR

spectrometer. The FTIR spectral readouts were collected at pH 7.5 immediately after dispensing the proteins in respective buffer solutions. Buffer baseline was subtracted before taking each spectrum. The deconvolution of raw spectra in the amide I region (1700 cm$^{-1}$ to 1600 cm$^{-1}$) was done using least-squares iterative curve fitting to Gaussian/Lorentzian line shapes. The assignment of peaks was done using previously described spectral components associated with different secondary structures (*Bandekar, 1992*). For investigating the morphological changes of bilayer due to interaction of protein variants and preformed aggregates, FTIR spectroscopy was utilized by using typical lipid concentration 2 mM. Background corrections were done for each and every experiment. We specifically evaluated the vibrational changes in the planar DPPC bilayer through the measurement of the changes in CH$_2$ wagging band frequency (1280–1460 cm$^{-1}$) of the hydrocarbon tail region was considered for evaluation.

## Unfolding transition of SOD1 mutants

Unfolding of mutant proteins on the basis of their tryptophan fluorescence was performed using guanidinium hydrochloride (GdmCl) by monitoring the quenching of tryptophan fluorescence of mutant proteins. A typical protein concentration of 10 µM was used for unfolding experiments. Experimental data were fit using the two-state transition model as follows:

$$y = \{(y_n + m_n x) + (y_n + m_d x)e^{-[-(\Delta G^0 + \frac{mx}{0.5826})]}\} / \{1 + e^{[-(\Delta G^0 + \frac{mx}{0.5826})]}\} \qquad (2)$$

where y denotes the observed fluorescence intensity, $y_n$ and $y_d$ refer to fluorescence signals at native and unfolded conditions, respectively, $\Delta G^0$ stands for free energy of unfolding transition, m is the cooperativity, and x denotes the concentration of GdmCl. All data analysis and fitting were carried out using OriginPro 8.5.

## AFM studies

Aliquots of aggregating samples were withdrawn after prolonged incubation at 37°C and were diluted with 5 mM phosphate buffer, pH 7.4. A 5 µL aliquot was taken from the diluted sample and deposited on freshly cleaved mica for 10 min. The typical protein concentration was taken 500 nM. After removing the excess liquid, the aggregates were rinsed with MilliQ water and then dried with a stream of nitrogen. Images were acquired at ambient temperature using a Bioscope Catalyst AFM (Bruker Corporation, Billerica, MA) with silicon probes. The standard tapping mode was used to image the morphology of aggregates. The nominal spring constant of the cantilever was kept at 20–80 N/m. The spring constant was calibrated by a thermal tuning method. A standard scan rate of 0.5 Hz with 512 samples per line was used for imaging the samples. A single third-order flattening of height images with a low pass filter was done followed by section analysis to determine the dimensions of aggregates.

## ThT fluorescence assay

WT SOD1, apo, H72F, and H121F proteins were subjected to mechanical agitation at 200 rpm at 37°C for 350 hr. The protein concentrations for the aggregate preparation were kept 50 µM in 20 mM sodium phosphate buffer at pH 7.5. The protein species were treated with 1.2 µM TCEP. For the measurement of membrane-induced protein aggregation, WT and other protein variants were incubated in lipid/protein ratio 2:1 under the treatment of TCEP. Aliquots were thereafter subjected to ThT addition and fluorescence measurements were taken using an integration time of 0.3 s. The steady-state fluorescence was monitored using an excitation wavelength of 450 nm, and the values of emission intensity at 485 nm were recorded.

## Assay for permeabilization of lipid vesicles

The ability of protein aggregates to promote the release of calcein from entrapped SUVs composed of DOPC:DOPE:DOPS in the ratio 2:5:3 was checked by monitoring the increase in fluorescence intensity of calcein. Calcein-loaded liposomes were separated from non-encapsulated (free) calcein by gel filtration on a Sephadex G-75 column (Sigma) using an elution buffer of 10 mM MOPS, 150 mM NaCl, and 5 mM EDTA (pH 7.4), and lipid concentrations were estimated by complexation with ammonium ferro-thiocyanate (*Stewart, 1980*). Fluorescence was measured at room temperature

(25°C) in a PTI spectro-fluorometer using a 1 cm path length cuvette. The excitation wavelength was 490 nm and emission was set at 520 nm. Excitation and emission slits with a nominal bandpass of 3 and 5 nm were used, respectively. The high concentration (10 mM) of the entrapped calcein led to self-quenching of its fluorescence resulting in low fluorescence intensity of the vesicles ($I_B$). Release of calcein caused by addition of proteins aggregates led to the dilution of the dye into the medium, which could therefore be monitored by an enhancement of fluorescence intensity ($I_F$). This enhancement of fluorescence is a measure of the extent of vesicle permeabilization. The experiments were normalized relative to the total fluorescence intensity ($I_T$) corresponding to the total release of calcein after complete disruption of all the vesicles by addition of Triton X-100 (2% v/v). The percentage of calcein release in the presence of different aggregates of SOD1 protein species was calculated using the equation (*Benachir et al., 1997*):

$$\% \, release = 100 \frac{(I_F - I_B)}{(I_T - I_B)} \tag{3}$$

where $I_B$ is the background (self-quenched) intensity of calcein encapsulated in vesicles, $I_F$ represents the enhanced fluorescence intensity resulting from the dilution of dye in the medium caused by protein-aggregate-induced release of entrapped calcein. $I_T$ is the total fluorescence intensity after complete permeabilization is achieved upon addition of Triton X-100. Typical SUV concentration was taken 100 μM, and 5 μM concentration of each protein aggregate sample was used. The pore formation rate constants (∧, cm-1) were calculated through the exponential fitting of the growth kinetics due to leakage of calcein from the entrapped SUVs.

## Preparation of GUVs

GUVs were formed in 0.1 M sucrose prepared in 1 mM HEPES (pH 7.4) buffer using electroformation, as described by *Pott et al., 2008*. Briefly, 20 μL of a 1 mM lipid (DOPC:DOPE:PI:DOPS:CL in the ratio 4.5:2.5:1:0.5:1.5) solution in chloroform was spread onto the surfaces of two conductive glasses (coated with Fluor Tin Oxide), which were then placed with their conductive sides facing each other. These droplets were allowed to dry overnight in a closed chamber containing saturated solution of NaCl. This is to avoid complete drying of the droplets. The hydration of these droplets facilitates electroformation process. Electroformation chamber was made using Teflon spacer of thickness ~2 mm. This electro swelling chamber was filled with 0.1 M sucrose solution and branched to an alternating power generator at 1.5 V and 15 Hz frequency during 2 hr at room temperature (22–25°C). The vesicles solution was then carefully transferred to an eppendorf vial and kept at rest at 4°C before use. The average diameter of the GUV obtained was 10–100 μm. For single GUV imaging, we selected the GUV of size ~30 μm for eliminating the effect of curvature. GUVs were diluted in 0.1 M glucose and prepared in 1 mM HEPES (pH 7.4), for observation. A typical observation experiment, using an inverted microscope, was made in an observation chamber by mixing 30 μL of the GUV solution with 100 μL of a 0.1 M glucose solution. The slight density difference between the inner and outer solutions drives the vesicles to settle at the bottom of the slide and provides better contrast while observing under phase contrast.

## Phase contrast optical microscopy

For the microscopy experiment, 5 μM concentrations of protein aggregates of different variants were dissolved in the glucose solution prepared in 1 mM HEPES buffer (pH 7.4) were added to the vesicle solution. Phase contrast microscopy was performed using an inverted microscope (DMi8) from Leica (Wetzlar, Germany). Observation chamber consisted of a glass slide with rubber spacers. The chamber was then closed immediately for observation under phase-contrast microscope after placing the samples. Response of individual GUV when exposed to the protein aggregates solution was continuously recorded with time using a CCD camera. Images were analyzed using the image analysis software, Image J. A straight line was drawn across the GUV to obtain an intensity profile. Peak to peak intensity ($I_{ptp}$) across the halo region was calculated. Average $I_{ptp}$ was obtained from several line profiles across the GUV. The time in second versus $I_{ptp}$ was plotted in order to observe any significant change in the intensity profile of the GUV. We have calculated the $I_{ptp}$ values for at least three different GUVs treated with different aggregate species and averaged the values to calculate the deformation rate constant ($\lambda$, sec$^{-1}$).

## Cell culture and cytotoxicity assay

Neuroblastoma cell lines SHSY5Y were acquired from the national cell repository (National Centre for Cell Science, Pune, India). Cells were authenticated using STR analysis. The cells tested negative for mycoplasma contamination as tested using PCR. Cells were maintained in Dulbecco's modified Eagle's media (DMEM), which in turn were supplemented with 10% heat-inactivated fetal bovine serum (FBS), respectively, 4.5 g/L of glucose, 1.5 g/L sodium bicarbonate, 110 mg/L sodium pyruvate, 4 mM L-glutamine, 50 units/ml penicillin G, and 50 µg/ml streptomycin in humidified air containing 5% $CO_2$ at 37°C. Sub-culturing was done by allowing the passaging of cells as per ATCC recommendations (ATCC, Manassus, VA). Cells were cultured in both serum and antibiotic free culture medium before each experiment. MTT assay was directed to evaluate the cell cytotoxicity (*Nandi et al., 2017*). For the initial screening experiment, the SHSY5Y cells ($4 \times 10^3$ cells per well) were seeded in a 96 well plate and left in an incubator followed by treatment with different protein aggregates variants (5 µM) for 12 hr. After 12 hr of incubation, cells were washed with PBS, and then the MTT solution was added to each well and kept in an incubator for 4 hr to form formazan salt. Then the formazan salt was solubilized using DMSO, and the absorbance was observed at 595 nm using an ELISA reader (Emax, Molecular Device).

## Labeling of the protein with Alexa488Maleimide

All the SOD1 protein species were labeled with Alexa488Maleimide (Alexa488) using a previously published procedure (*Kundu et al., 2017*). Briefly, Alexa488 dissolved in DMSO was slowly added to 2 mg mL$^{-1}$ solution of the protein under constant stirring. The molar ratio between the dye and protein was maintained at 10:1. The reaction mixture was incubated at 4°C for 5 hr, with shaking after every 30 min. The labeling reaction was then stopped by adding excess β-mercaptoethanol. Excess free dye from the reaction mixture was removed by extensive dialysis followed by column chromatography using a Sephadex G20 column which was pre-equilibrated with 20 mM sodium phosphate buffer (pH 7.5).

## FCS experiments and data analysis

FCS experiments were carried out using a dual channel ISS Alba V system equipped with a 60× water-immersion objective (NA 1.2). Samples were excited with an argon laser at 488 nm. All protein data were normalized using the $\tau_D$ value obtained with the free dye (Alexa488), which was measured under identical conditions. For a single-component system, diffusion time ($\tau_D$) of a fluorophore and the average number of particles (N) in the observation volume can be calculated by fitting the correlation function [$G(\tau)$] to *Equation4*:

$$G(\tau) = 1 + \left( \frac{1}{N(1+\frac{\tau}{\tau D})} \right) \frac{1}{\sqrt{1+S^2 \frac{\tau}{\tau D}}} \tag{4}$$

where S is the structure parameter, which is the depth-to-diameter ratio. The characteristic diffusion coefficient (D) of the molecule can be calculated from $\tau_D$ using *Equation 5*:

$$\tau D = \frac{\omega^2}{4D} \tag{5}$$

where $\omega$ is the radius of the observation volume, which can be obtained by measuring the $\tau_D$ of a fluorophore with known D value. The value of hydrodynamic radius ($r_H$) of a labeled molecule can be calculated from D using the Stokes–Einstein equation (*Equation 6*):

$$D = \frac{kT}{6\pi\eta rH} \tag{6}$$

where, k is the Boltzmann constant, T is the temperature, and $\eta$ corresponds to the viscosity of the solution (*Chattopadhyay et al., 2002*).

## WSME model

We employ the Ising-like WSME model (*Wako and Saitô, 1978*; *Muñoz and Eaton, 1999*) with the block approximation (*Gopi et al., 2019*) to predict the conformational landscape of SOD1 oxidized

monomer and its variants using the PDB structure 4FF9 as the reference. Briefly, the model assigns binary variables of *1* or *0* for folded or unfolded status of residues, respectively. We employ a version of the model that accounts of single-stretches of folded blocks (single-sequence approximation), two stretches of folded blocks (double-sequence approximation [DSA]) and DSA allowing for interactions across the folded islands if they are interacting in the folded structure. The 151-residue protein SOD1 is therefore reduced to a collection of 49 sequential blocks on assuming a block length of 3 thus reducing the number of microstates from >42,700,000 (in the residue-level version of the model) to 461,826. The energetics of the model involves van der Waals interactions identified with a 6 Å heavy-atom cut-off, all-to-all Debye–Hückel electrostatics, and simplified solvation (defined by the heat capacity change per native contact of) (*Naganathan, 2012*). Residues identified to be fully folded are assigned an entropic penalty of −13.6 J mol$^{-1}$ K$^{-1}$ per residue ($\Delta S_{conf}$). The apo form of SOD1 is simulated by assigning an excess conformational entropy of −19.7 J mol$^{-1}$ K$^{-1}$ per residue (*Rajasekaran et al., 2016*) for the stretches of residues 49–82 (loop IV, Zn binding loop) and 121–142 (loop VII, Cu binding loop), as reported from NMR order parameter measurements (*Sekhar et al., 2015*). To simulate order in either one or both the loops, the conformational entropy of residues in the loop is modified to −13.6 J mol$^{-1}$ K$^{-1}$ per residue (i.e. a lower penalty for folding), thus mimicking the variants of SOD1 (Zn bound, Cu bound, and Holo forms). The van der Waals interaction energies are fixed to −35.9, –38.2, −42.2, and −48.9 J mol$^{-1}$ for the Holo, Zn bound, Cu bound, and apo variants, respectively, to simulate iso-stability conditions at 298 K. The heat capacity change per native contact is fixed to −0.36 J mol$^{-1}$ K$^{-1}$ per native contact. All prolines are assigned an entropic penalty of zero to account for their rigidity. Residue probabilities and folding mechanism as a function of the number of structured blocks are predicted at iso-stability conditions (i.e. a stability 25 kJ mol$^{-1}$ at 298 K) following established protocols by accumulating partial partition functions.

## Orientations of proteins in membranes

To gain an insight as to how the WT and metal starved variants of SOD1 interact with membrane, we resorted to computational approaches. Protein orientations in membranes were theoretically calculated by minimizing a protein's transfer energy from water to a planar slab that serves as a crude approximation of the membrane hydrocarbon core. For WT SOD1, we referred to the solved structure 4BCY and for the metal starved forms in vacuo ab-initio models were prepared from Zhang Lab server (*Yang and Zhang, 2015*). The membrane binding propensity was calculated submitting the co-ordinate information of the protein forms to OPM server. A protein was considered as a rigid body that freely floats in the planar hydrocarbon core of a lipid bilayer. Accessible surface area is calculated using the subroutine SOLVA from NACCESS with radii of Chothia and without hydrogen (*Lomize et al., 2006*; *Lomize et al., 2007*). In OPM, solvation parameters are derived specifically for lipid bilayers and normalized by the effective concentration of water, which changes gradually along the bilayer normal in a relatively narrow region between the lipid head group regions and the hydrocarbon core.

## Statistical methods

Unless otherwise noted, all p-values were calculated by performing a paired or unpaired t-test.

## Acknowledgements

Author A Sannigrahi acknowledges University Grant Commission, Govt. of India for providing senior research fellowship. S Karmakar acknowledges the financial support from DBT-funded research project (BT/PR8475/BRB/10/1248/2013). K Chattopadhyay acknowledges funding from Science and Engineering Research Board (SERB) (Fund Number: PCR_EMR_2016_000310). We thank the central instrument facility (CIF) for their help in multiple experiments, including FT-IR, CD, dynamic light scattering, FCS among other. We thank the director, CSIR-IICB for help and encouragements.

## Additional information

### Funding

| Funder | Grant reference number | Author |
|---|---|---|
| University Grants Commission | Fellowship | Achinta Sannigrahi<br>Sourav Chowdhury |
| Department of Biotechnology , Ministry of Science and Technology | BT/PR8475/BRB/10/1248/2013 | Sanat Karmakar |
| Science and Engineering Research Board | EMR/2016/000310 | Krishnananda Chattopadhyay |
| CSIR | CSIR internal funding | Krishnananda Chattopadhyay |

The funders had no role in study design, data collection and interpretation, or the decision to submit the work for publication.

### Author contributions

Achinta Sannigrahi, Sourav Chowdhury, Conceptualization, Formal analysis, Investigation, Methodology, Writing - original draft; Bidisha Das, Formal analysis, Investigation, Writing - review and editing; Amrita Banerjee, Animesh Halder, Mohammed Saleem, Investigation; Amaresh Kumar, Methodology; Athi N Naganathan, Formal analysis, Methodology, Writing - review and editing; Sanat Karmakar, Formal analysis, Funding acquisition, Methodology, Writing - original draft; Krishnananda Chattopadhyay, Conceptualization, Funding acquisition, Writing - original draft, Project administration, Writing - review and editing

### Author ORCIDs

Achinta Sannigrahi (iD) https://orcid.org/0000-0002-6037-2805
Athi N Naganathan (iD) http://orcid.org/0000-0002-1655-7802
Sanat Karmakar (iD) http://orcid.org/0000-0001-7627-8904
Krishnananda Chattopadhyay (iD) https://orcid.org/0000-0002-1449-8909

### Decision letter and Author response

Decision letter https://doi.org/10.7554/eLife.61453.sa1
Author response https://doi.org/10.7554/eLife.61453.sa2

## Additional files

### Supplementary files

• Transparent reporting form

### Data availability

All data generated or analyzed during this study are included in the manuscript and supporting files.

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
