## [Decision Letter]

**Acceptance summary:**

Mutation of the superoxide dismutase is implicated in motor neuron disease. The enzyme contains both zinc (Zn) and copper (Cu) as cofactors. The role that either Zn or Cu play in membrane association and disease-causing aggregation of superoxide dismutase is currently unclear. The authors apply site-directed mutagenesis to generate Zn-only and Cu-only binding mutants of the enzyme and untangle the effect that the binding of each cofactor has on membrane association and aggregation. Through application of a large set of complementary techniques, involving statistical mechanical modelling, fluorescence and infra-red spectroscopy, and optical and atomic force microscopy, the authors show that deficiency of Zn uptake, induced by mutation, is a major driving force for toxic aggregation of superoxide dismutase.

**Decision letter after peer review:**

Thank you for submitting your article "Metal cofactor zinc and interacting membranes modulate SOD1 conformation-aggregation landscape in an in vitro ALS Model" for consideration by *eLife*. Your article has been reviewed by 3 peer reviewers, one of whom is a member of our Board of Reviewing Editors, and the evaluation has been overseen by Olga Boudker as the Senior Editor. The following individual involved in review of your submission has agreed to reveal their identity: Erdinc Sezgin (Reviewer #2).

The reviewers have discussed the reviews with one another. They acknowledge the integrated approach taken by you and your co-authors and the amount of data presented and discussed. However, the reviewers raise major concerns regarding both experiments and computer simulations. Not all conclusions are justified by the data presented and additional data are required. We ask you to revise your manuscript in light of the reviewers' concerns.

Reviewer #1:

Sannigrahi et al. report the investigation of structural determinants of membrane insertion and aggregation of Cu-Zn superoxide dismutase (SOD1), an enzyme that is implicated in motor neuron disease. The authors combine mutagenesis experiments with a variety of techniques, involving tryptophan fluorescence, FTIR, AFM, Tht fluorescence, FCS, optical microscopy and computer simulation. They arrive at that conclusion that conformational change and site-specific metal binding modulate membrane insertion and aggregation of SOD1.

Identifying the origins of SOD1 dysfunction and aggregation can have important implications in the development of therapeutic strategies for motor neuron disease. The underlying molecular biology is not well understood. The study by Sannigrahi et al. is an integrated approach involving an impressive number of complementary methods. However, the conclusions put forward are not sufficiently supported by the data presented. The applied methodologies yield data of insufficient resolution to draw the detailed molecular picture presented. Additional experimental work would be required to substantiate or provide evidence for the findings.

1. The statistical mechanical model (WSME) is coarse-grained. It e.g. considers three consecutive amino acid residues as a block. It is therefore of limited suitability to study the effects of single-point mutations and metal-binding or conformation and aggregation.

2. The effect of mutation and Zn/Cu-binding on Trp fluorescence spectral properties of SOD1 is marginal (Figure 2a). Likewise, the far-UV CD spectra shown in supporting information show marginal changes. The broad spectral characteristics of far-UV CD defies an accurate, quantitative deconvolution of secondary structure content. No solid conclusions concerning a conformational change can thus be inferred. FTIR spectra are broad and smooth (i.e. lack significant sub-structure) (Figure 2b, c). Their deconvolution in seven discrete sub-states appears ambitious and error-prone.

3. The authors propose to determine membrane affinities of SOD1 and mutants thereof by applying extrinsic fluorescence modification and by measuring binding to artificial micelles using fluorescence correlation spectroscopy (analysis of diffusion time constants). Extrinsic fluorescence labels are hydrophobic compounds and supposedly tend to strongly interact with membrane lipids. This will provides an artificial bias of conjugates to micelle membranes. Control experiments are required to rule out effects of the labels.

4. The influence of mutation on stability and conformation of SOD1 is unclear. Mutations H72F and H121F, introduced to alter metal binding, may as well have effects on stability and conformation (folding) of the entire domain, irrespective of the metal-bound/unbound state. Mutation itself may lead to unfolding and aggregation. Mutation of a histidine to a phenylalanine, as applied by the authors, may have disruptive effects on protein structure because a small side chain is replaced by a larger one. Thermal and/or chemical denaturation experiments, carried out on isolated protein material and mutants thereof, and their analysis are required to assess the effect of mutations on folding and stability.

Reviewer #2:

In this manuscript, Sannigrahi et al. studied the role of metal binding sites of SOD1 on its aggregation and toxicity. They created a Zn only, Cu only binding mutants as well as Zn/Cu binding-deficient mutant. Zn bearing mutant behaved similarly as wild type protein in terms of membrane binding, aggregate formation and toxicity, while Zn/Cu deficient mutant behaved similarly to Cu bearing (no Zn) mutant. They conclude that Zn binding pocket is crucial to keep the protein in healthy state and in the absence of Zn binding, protein aggregates especially in the presence of membranes. Lastly, they investigated real disease mutations and samples two mutations with different degree of Zn binding, and confirmed the same trend; if the Zn binding pocket is influenced, mutation is more severe.

I am not an expert of this particular biological question (ALS and role of SOD1), but I evaluated the technical aspects of the manuscript.

In general, the manuscript is well written, the messages are clear and the conclusions are supported by data.

Reviewer #3:

This paper looks at the effect of metal cofactor binding on the aggregation and toxicity of SOD1, which natively binds a Cu2+ and a Zn2+ ion. The authors investigate the WT SOD1, the apo SOD1 and two mutants which do not bind Cu2+ (H121F) or Zn2+ (H72F) in order to look at the effects of the metal binding on SOD aggregation and toxicity. They find by a number of assays and a computational study that Zn2+ rather than Cu2+ is the dominant factor in determining susceptibility to aggregation, membrane binding, etc. Based on this they propose that deficient Zn2+ uptake by SOD1 is responsible for the pathogenic behaviour of some mutants.

There is a lot of interesting data in this paper supporting this hypothesis (some more so than others), however there are some points the authors should consider:

1. A potential weakness of the computational estimation of membrane binding affinity is that the WT crystal structure was used for WT, while structure predictions from the I-TASSER server were used for apo and Cu/Zn-deficient mutants. Since one might expect the predicted structure to be of lower quality, it might then have an enhanced propensity for membrane binding via exposed hydrophobic groups? What would be obtained if the I-TASSER server was also used to generate the structure used for WT in this calculation? This point also applies to the computational validation where predicted membrane binding free energies are compared with distance to the Zn2+ or Cu2+ site of the mutants. This again involves a 2-stage prediction – firstly of the mutant structure, then of its binding energy. Maybe the authors can give some intuition as to how this can be sufficiently accurate to be useful?

2. Correlation functions for A488-SOD1 are shown are at the extremes of no SUVs versus a high concentration of SUVs. What happens at intermediate concentrations where there would be more of a mix of bound and unbound populations – can the two components be clearly resolved in the log-linear plots of G(tau)?

3. I may have missed something, but why does the population of membrane-bound protein saturate at much less than 100%? Is there a baseline parameter for the population at high [DPPC SUV] in addition to Ka? One thing that occurred to me is that membrane binding may quench the fluorescence somewhat, so the amplitude of the membrane-bound population may be lower than it should be, hence this effect; and the differences in folding/misfolding of the SOD mutants may lead to different binding to the SUVs which would in turn affect the relative amplitudes of the two components. This wouldn't affect the fit of the sigmoidal curves, but maybe the relative fraction of slowly diffusing component should not be literally interpreted in terms of a bound population. Rather than "population membrane bound" Figure 2f could say "Fraction bound fluorescence" or similar? This interpretation would support the authors' contention that H72F is more apo-like and H121F more holo-like.

4. The differences in the ratio Ksvm/Ksv are basically reflecting differences in Ksv, because the values of Ksvm are all very similar. Thus it may reflect more the differences in non membrane-bound protein than differences in membrane binding, as seems to be the inference in the paper?

5. The finding of change in secondary structure on membrane binding based on IR data, in particular increase in alpha-helical population, for the apo form and the H72F, is very interesting and strongly supports differences in membrane interaction between WT/H121F and apo/H72F – maybe this data should be included in the main text rather than the SI in fact? To me this seems a more noteworthy change than the modest differences in membrane association constants obtained from FCS.

6. Aggregation was studied for the reduced form of the disulfides. The authors should motivate why the aggregation is studies using the reduced form of the protein while the prior work in the paper used the oxidized form (I believe?). My knowledge in this area is limited so I'm not sure which is the form more relevant to observed pathologies.

7. A complicating factor in the perturbation of GUV membranes by the aggregates formed with/without SUVs present is the SUVs themselves. Presumably there is a significant SUV concentration in the aliquots taken from the aggregation reaction – could the SUVs rather than differences in the aggregates be responsible for the difference in the effect on GUVs? A control could be to add just SUVs to the GUV samples.

8. For the validation, a statistical test should be used to demonstrate the significance of the observed correlations.

---

## [Author Response]

Reviewer #1:Sannigrahi et al. report the investigation of structural determinants of membrane insertion and aggregation of Cu-Zn superoxide dismutase (SOD1), an enzyme that is implicated in motor neuron disease. The authors combine mutagenesis experiments with a variety of techniques, involving tryptophan fluorescence, FTIR, AFM, Tht fluorescence, FCS, optical microscopy and computer simulation. They arrive at that conclusion that conformational change and site-specific metal binding modulate membrane insertion and aggregation of SOD1.Identifying the origins of SOD1 dysfunction and aggregation can have important implications in the development of therapeutic strategies for motor neuron disease. The underlying molecular biology is not well understood. The study by Sannigrahi et al. is an integrated approach involving an impressive number of complementary methods. However, the conclusions put forward are not sufficiently supported by the data presented. The applied methodologies yield data of insufficient resolution to draw the detailed molecular picture presented. Additional experimental work would be required to substantiate or provide evidence for the findings.

We thank the reviewer for the appreciation of our work. We appreciate the careful and detailed comments by the reviewer, and we have included additional details (including new data and characterization), which we believe significantly improves the clarity and robustness of our results.

1. The statistical mechanical model (WSME) is coarse-grained. It e.g. considers three consecutive amino acid residues as a block. It is therefore of limited suitability to study the effects of single-point mutations and metal-binding or conformation and aggregation.

The reviewer’s comment is well-taken. The WSME model in its current version is not suitable to study the role of single-point mutations. We could use the conventional version with every residue as a folding unit but that increases the number of conformations to 42.7 million, compared to ~450,000 in the 3-residue block representation. In order to not over interpret we have used the WSME model as a starting point to explore the folding landscape of SOD1 and the differences in folding mechanism, if any, from the model by simulating the presence of different ions and in their absence – note that this simulation derives directly from NMR order parameters and hence constrained by experiment on the regions of the protein that rigidify on ion-binding. These predictions are then tested directly via experiments in the rest of the manuscript. We therefore do not use the model to study aggregation or the effect of single point mutations on aggregation, but only to obtain initial insights into the folding conformational landscape, which can be verified by experiments.

2. The effect of mutation and Zn/Cu-binding on Trp fluorescence spectral properties of SOD1 is marginal (Figure 2a). Likewise, the far-UV CD spectra shown in supporting information show marginal changes. The broad spectral characteristics of far-UV CD defies an accurate, quantitative deconvolution of secondary structure content. No solid conclusions concerning a conformational change can thus be inferred. FTIR spectra are broad and smooth (i.e. lack significant sub-structure) (Figure 2b, c). Their deconvolution in seven discrete sub-states appears ambitious and error-prone.

We thank the reviewer. We agree with the reviewer that the change is small between mutant proteins, when a particular method is considered, like fluorescence or CD. This is exactly why we used a combination of multiple techniques to support our inference. For example, we agree with the reviewer that the change in tryptophan fluorescence spectra of the mutants is small (Figure 2a). However, several repeat experiments (three different experiments using freshly prepared samples with three repeats for each experiment) confirmed that while the change is small, it is reproducible. We substantiated the change in steady state fluorescence experiments with acrylamide quenching measurements, which showed reproducible and significant change in the solvent surface exposure of the single tryptophan residue of the mutants (Table 2, Figure 2—figure supplement 3).

To avoid over-interpretation, where we discussed the steady state fluorescence measurements, we only mentioned the cases of WT and apo, where the changes are significantly more. The two mutants were mentioned for their change in Ksv values. Please refer to the following sentences at the Result section of the manuscript:

“Steady state fluorescence maxima in combination with acrylamide quenching data suggested an appreciable conformational alteration in going from the WT to the apo form. Interestingly, Zn-starved H72F mutant showed higher *K_sv_* compared to Cu-starved H121F mutant (Figure 2—figure supplement 3, Table 2).”

Similarly, while the change in far UV CD in our measurements is small, it follows the trends observed by other techniques. It may be noted that we did not use the CD data for the estimation of the secondary structure contents. For a complimentary measurement of the CD, and to provide an estimation of the secondary structure contents, we used FT-IR data. We apologize as we have not provided in the previous version sufficient information about the analyses of the FT-IR data, which is now supplied. We have analyzed FT-IR data using two steps. First, we have determined the second derivatives of the FT-IR data (Figure 2—figure supplement 5). The peak regions were chosen from the second derivatives. In the second step, we used these peak positions to fit the original FT-IR data to calculate the percentage of the secondary structures. It may be noted that this method of fitting FT-IR data is already published (DOI:10.1111/j.17457270.2007.00320.x;DOI:10.1111/j.174570.2007.00320.x,DOI:10.1038/nprot.2015.0 24). Also, the percentage of the secondary structure determined for WT SOD1 using this method matches well with data published by others with the same protein (DOI: 10.1111/j.17457270.2007.00320.x, DOI: 10.1039/d0cb00203h).

We reemphasize that while the change in a particular spectroscopic property is small but reproducible for protein variants between these two groups (apo, Zn deficient mutant vs WT, Cu deficient mutant), the trend has been found identical when we used different techniques (like steady state fluorescence emission, fluorescence quenching, far UV CD and FT-IR). In addition, the trend continues to remain valid when we proceed through the rest of the paper (for example, membrane binding, aggregation propensity etc).

3. The authors propose to determine membrane affinities of SOD1 and mutants thereof by applying extrinsic fluorescence modification and by measuring binding to artificial micelles using fluorescence correlation spectroscopy (analysis of diffusion time constants). Extrinsic fluorescence labels are hydrophobic compounds and supposedly tend to strongly interact with membrane lipids. This will provides an artificial bias of conjugates to micelle membranes. Control experiments are required to rule out effects of the labels.

We thank the reviewer for suggesting control experiments. In the previous version, we have provided data of FCS experiments to determine the binding constants of the alexa-488 maleimide labeled protein variants with DPPC vesicles. This is a standard method of estimation of binding constants between protein and lipid vesicles as reported earlier

(doi: 10.1016/j.bpj.2010.07.056). We assumed that since the label is identical for all protein variants, the change induced by the label will be similar for all variants. However, we do agree with the concerns raised by the reviewer. To address the reviewer’s concern and to provide complementary data, we have now measured the tryptophan fluorescence of SOD1 variants with increasing concentrations of DPPC SUVs. From the gradual enhancement of tryptophan fluorescence due to lipid binding, we calculated the binding affinities of all protein variants towards membrane which showed comparable binding constants as obtained from our FCS experiments. We have added additional figures as Figure 2—figure supplement 7b,c.

4. The influence of mutation on stability and conformation of SOD1 is unclear. Mutations H72F and H121F, introduced to alter metal binding, may as well have effects on stability and conformation (folding) of the entire domain, irrespective of the metal-bound/unbound state. Mutation itself may lead to unfolding and aggregation. Mutation of a histidine to a phenylalanine, as applied by the authors, may have disruptive effects on protein structure because a small side chain is replaced by a larger one. Thermal and/or chemical denaturation experiments, carried out on isolated protein material and mutants thereof, and their analysis are required to assess the effect of mutations on folding and stability.

We thank the reviewer for the comment and suggestion. As per reviewer’s suggestion, we performed the unfolding transition experiments using guanidine hydrochloride.

We don’t see significant change in case of H121F and H72F (Figure 2—figure supplement 2).

Reviewer #3:This paper looks at the effect of metal cofactor binding on the aggregation and toxicity of SOD1, which natively binds a Cu2+ and a Zn2+ ion. The authors investigate the WT SOD1, the apo SOD1 and two mutants which do not bind Cu2+ (H121F) or Zn2+ (H72F) in order to look at the effects of the metal binding on SOD aggregation and toxicity. They find by a number of assays and a computational study that Zn2+ rather than Cu2+ is the dominant factor in determining susceptibility to aggregation, membrane binding, etc. Based on this they propose that deficient Zn2+ uptake by SOD1 is responsible for the pathogenic behaviour of some mutants.There is a lot of interesting data in this paper supporting this hypothesis (some more so than others), however there are some points the authors should consider:1. A potential weakness of the computational estimation of membrane binding affinity is that the WT crystal structure was used for WT, while structure predictions from the I-TASSER server were used for apo and Cu/Zn-deficient mutants. Since one might expect the predicted structure to be of lower quality, it might then have an enhanced propensity for membrane binding via exposed hydrophobic groups? What would be obtained if the I-TASSER server was also used to generate the structure used for WT in this calculation? This point also applies to the computational validation where predicted membrane binding free energies are compared with distance to the Zn2+ or Cu2+ site of the mutants. This again involves a 2-stage prediction – firstly of the mutant structure, then of its binding energy. Maybe the authors can give some intuition as to how this can be sufficiently accurate to be useful?

We thank the reviewer for the comments on computational estimate of membrane binding affinity. As per reviewer suggestion, we constructed the WT SOD1 model structure from ITASSER and calculated the binding affinity of model structure with membrane. Our calculated free energy (ΔG) of binding was appeared to be -1.2 Kcal/mole which is closer to the value which was obtained using the WT crystal structure. We have modified the binding affinity in the revised version of the manuscript. On the other hand, we have used all the available structures (crystal/NMR) of the disease mutants and calculated the binding free energies and Zn2+ or Cu2+ site distances from mutational location as well and plotted their respective free energy values against the distance parameters (Figure 5).

2. Correlation functions for A488-SOD1 are shown are at the extremes of no SUVs versus a high concentration of SUVs. What happens at intermediate concentrations where there would be more of a mix of bound and unbound populations – can the two components be clearly resolved in the log-linear plots of G(tau)?

We thank the reviewer for the valued comment. We are now providing data of correlation functions for an intermediate concentration. Please refer to Figure 2g in the revised manuscript version which showed the existence of two components in the correlation curve.

3. I may have missed something, but why does the population of membrane-bound protein saturate at much less than 100%? Is there a baseline parameter for the population at high [DPPC SUV] in addition to Ka? One thing that occurred to me is that membrane binding may quench the fluorescence somewhat, so the amplitude of the membrane-bound population may be lower than it should be, hence this effect; and the differences in folding/misfolding of the SOD mutants may lead to different binding to the SUVs which would in turn affect the relative amplitudes of the two components. This wouldn't affect the fit of the sigmoidal curves, but maybe the relative fraction of slowly diffusing component should not be literally interpreted in terms of a bound population. Rather than "population membrane bound" Figure 2f could say "Fraction bound fluorescence" or similar? This interpretation would support the authors' contention that H72F is more apo-like and H121F more holo-like.

We thank the reviewer for this valuable comment. The percentage was calculated using the two components fit of the correlation function. We apologize that there is an error the way we calculated the percentages. We corrected this in the new version of the manuscript in Figure 2i.

We have followed the procedure of Middleton et al. for the calculation of the percentages of the bound/free SOD1(E. R. Middleton and E. Rhoades, Biophysical journal, 2010, 99, 2279-2288.). As per reviewer’s suggestion we have replaced the term ‘population membrane bound’ by "Fraction bound’’ in the modified figure in the revised manuscript version.

4. The differences in the ratio Ksvm/Ksv are basically reflecting differences in Ksv, because the values of Ksvm are all very similar. Thus it may reflect more the differences in non membrane-bound protein than differences in membrane binding, as seems to be the inference in the paper?

We agree. We have modified the portion slightly and mentioned how this ratio changed for different protein variants (without inferring anything from here).

5. The finding of change in secondary structure on membrane binding based on IR data, in particular increase in alpha-helical population, for the apo form and the H72F, is very interesting and strongly supports differences in membrane interaction between WT/H121F and apo/H72F – maybe this data should be included in the main text rather than the SI in fact? To me this seems a more noteworthy change than the modest differences in membrane association constants obtained from FCS.

We thank the reviewer for the valuable suggestion. We have now moved the membrane binding based on IR data, in particular increase in alpha-helical population to the main text from the SI. Please refer to Figure 2j,k in the revised manuscript version.

6. Aggregation was studied for the reduced form of the disulfides. The authors should motivate why the aggregation is studies using the reduced form of the protein while the prior work in the paper used the oxidized form (I believe?). My knowledge in this area is limited so I'm not sure which is the form more relevant to observed pathologies.

We thank the reviewer for the comment. In its mature form, SOD1 is a highly stable, homodimeric protein, with each subunit binding one catalytic copper ion and one structural zinc ion, and containing one intramolecular disulfide bond as well as two non-conserved free cysteines. Numerous in vivo and in vitro studies have shown that various immature, destabilized forms of SOD1 are prone to aggregate, and this is often enhanced by disease-associated mutations (Stathopulos PB, et al. 2003. Proc Natl Acad Sci USA 100:7021–7026; Karch CM, Prudencio M, Winkler DD, Hart PJ, Borchelt DR 2009Proc Natl Acad Sci USA 106:7774– 7779). Recently, much attentions has been focused on aggregation of the most immature form of SOD1, in which the disulfide bonds are reduced (DOI:10.1021/acschemneuro.7b00162). Studies of various mutant-SOD1 ALS mice models have shown that small, soluble, misfolded forms of reduced apo SOD1 are enriched in the spinal cord and may be the common cytotoxic species that cause ALS (Wang J, et al. (2009) Proc Natl Acad Sci USA 106:1392–1397; Zetterstrom P, et al. (2007) Proc Natl Acad Sci USA 104:14157–14162). In addition, cell culture studies suggest that ALS associated mutations can promote disulfide bond reduction and metal loss (Tiwari A, Hayward LJ (2003) Familial amyotrophic lateral sclerosis mutants of copper/zinc superoxide dismutase are susceptible to disulfide reduction. J Biol Chem 278:5984–5992.). Relatively little is known, though, about the properties of reduced apo SOD1, and how mutations affect these properties. Therefore, in this work we investigated in details the aggregation and toxicity of SOD1 protein variants in their disulfide reduced form.

7. A complicating factor in the perturbation of GUV membranes by the aggregates formed with/without SUVs present is the SUVs themselves. Presumably there is a significant SUV concentration in the aliquots taken from the aggregation reaction – could the SUVs rather than differences in the aggregates be responsible for the difference in the effect on GUVs? A control could be to add just SUVs to the GUV samples.8. For the validation, a statistical test should be used to demonstrate the significance of the observed correlations.

We thank the reviewer for the comment and suggestion. We performed the control GUV experiment using 10 μM of DPPC SUVs and calculated the Iptp values with time. Our results showed insignificant changes in the value of Iptp. This result has been provided as an additional figure as Figure 4—figure supplement 5 in the revised manuscript.